# A genome-wide association study in 10,000 individuals links plasma N-glycome to liver disease and anti-inflammatory proteins

Sodbo Sharapov [1], Anna Timoshchuk [1], Olga Zaytseva[2], Denis E. Maslov [1], Anna Soplenkova [1], Elizaveta E. Elgaeva [3,4], Evgeny S. Tiys[3], Massimo Mangino [5,6], Clemens Wittenbecher[7], Lennart Karssen [8], Maria Timofeeva[9,10], Arina Nostaeva[3,8], Frano Vuckovic [2], Irena Trbojević-Akmačić [2], Tamara Štambuk[2], Sofya Feoktistova[3], Nadezhda A. Potapova[11], Viktoria Voroshilova[3,12], Frances Williams [5], Dragan Primorac [13,14,15], Jan Van Zundert [16,17], Michel Georges[18], Karsten Suhre [19], Massimo Allegri [20], Nishi Chaturvedi [21], Malcolm Dunlop[9], Matthias B. Schulze [22,23,24], Tim Spector [5], Yakov A. Tsepilov [3,25], Gordan Lauc [2,26] & Yurii S. Aulchenko [1,3] ✉

More than a half of plasma proteins are N-glycosylated. Most of them are synthesized, glycosylated, and secreted to the bloodstream by liver and lymphoid tissues. While associations with N-glycosylation are implicated in the rising number of liver, cardiometabolic, and immune diseases, little is known about the genetic regulation of this process. Here, we performed the largest genome-wide association study of N-glycosylation of the blood plasma proteome in 10,000 individuals. We doubled the number of genetic loci known to be associated with blood N-glycosylation by identifying 16 novel loci and prioritizing 13 novel genes contributing to N-glycosylation. Among these were the *GCKR*, *TRIB1*, *HP, SERPINA1* and *CFH* genes. These genes are predominantly expressed in the liver and show a previously unknown genetic link between plasma protein N-glycosylation, metabolic and liver diseases, and inflammatory response. By integrating glycomics, proteomics, transcriptomics, and genomics, we provide a resource that facilitates deeper exploration of disease pathogenesis and supports the discovery of glycan-based biomarkers.

During maturation, more than half of human proteins are modified by the covalent linking of complex carbohydrates – glycans[1]. Glycoproteins comprise various secreted and membrane enzymes, receptors, hormones, cytokines, immunoglobulins, as well as structural and adhesion molecules[2]. Glycans affect the physical and chemical properties of proteins and their biological function[3–6]. Adequate glycosylation is required for the normal physiological action of glycoproteins, while aberrant glycosylation is increasingly implicated in human diseases[7–9]. Glycans are considered to be potential therapeutic targets[10–12], an essential part of therapeutics[13–15], as well as biomarkers for cancer, diabetes, inflammatory and autoimmune disease[16–18], which makes glycobiology a promising field for future clinical applications.

N-glycosylation is the most abundant type of glycosylation[1] and, unlike other types, is specific to a consensus asparagine-containing sequence (Asn-X-Ser/Thr, where X is any amino acid except Pro) in the protein's primary structure. Human N-glycans are irregular branched polymers consisting of mannose, galactose, fucose, sialic acid, and *N*-acetylglucosamine (GlcNAc) residuals, whose combinations introduce

a great diversity of protein glycoforms. Unlike proteins, whose primary structure is encoded in the genomic DNA sequence, the occupancy of the N-glycosylation site and the abundance of specific N-glycan structures are not directly encoded in the human genome. Protein glycosylation depends on the interplay of multiple enzymes catalyzing glycan transfer, glycosidic linkage hydrolysis, and glycan biosynthesis. The abundance of specific protein glycoforms can be influenced by various parameters, including the activity of enzymes and availability of substrates, the accessibility of a glycosylation site, protein synthesis, and degradation[19]. Overall, protein glycosylation is a complex process controlled by genetic, epigenetic, and environmental factors[20–22].

While the biochemical network of human N-glycan biosynthesis is well understood[23], little is known about in vivo regulation of this process[24], including tissue- and protein-specific regulation. A major part of the plasma glycoproteins consists of immunoglobulins, produced by antibody-producing B-cells, and secreted proteins produced in the liver[25]. Therefore, the N-glycosylation of blood plasma proteins serves as an indicator of liver and B-cell function. Study of plasma protein N-glycosylation potentially provides insights into the etiology and pathophysiology of liver and B-cell-mediated diseases, as well as diseases where these tissues are important players, such as cardiometabolic diseases and inflammatory conditions. Understanding the mechanisms underlying blood plasma glycosylation and its regulation at the tissue-specific level is crucial for unraveling the complex interplay between protein modifications, cellular functions, and disease processes.

In this context, genetics offers an attractive approach to studying the regulation of N-glycosylation in vivo and sheds light on how these molecular phenotypes are linked to human disease[25,26]. By the beginning of the 2010s, advancements in high-throughput methods for N-glycome profiling facilitated the genetic control of N-glycosylation[27]. As for other quantitative phenotypes, the genome-wide association study (GWAS) and multivariate genetic association analysis[28,29] may be applied to N-glycans to identify genetic loci associated with abundance and, therefore, contain genes involved in the regulation of N-glycosylation. Further integration of N-glycome GWAS results with other layers of biological information (e.g., biological pathways, protein-protein interactions, transcriptomics, proteomics, and others) allows the discovery of novel candidate genes regulating this process and provides hypotheses about biological mechanisms underlying the genetic associations[30,31]. A joint analysis of GWAS results of N-glycome and disease (e.g., pleiotropy analysis[32] and analysis of causal relationships using Mendelian randomization[33]) can shed light on how protein glycosylation is involved in the pathogenesis of human disease and suggest possible glycome-based biomarkers. Previous GWAS of total plasma N-glycome[34–37] identified 15 genetic loci and suggested the role of 19 candidate genes. These studies were supplemented with GWAS of N-glycome of immunoglobulin G (IgG)[28,38–41] and transferrin (TF)[42] glycoproteins, identifying an additional 19 loci and prioritizing 26 candidate genes[25]. The role of three candidate genes, encoding transcriptional factors *HNF1A*, *IKZF1*, and *RUNX3*, in the regulation of N-glycosylation was experimentally confirmed in vitro[34,40]. A Mendelian randomization study of IgG N-glycome found that the abundance of N-glycans with bisecting GlcNAc is a potential biomarker of systemic lupus erythematosus[43]. However, there remains a limited understanding of the role of gene regulators of N-glycosylation in health and disease.

The first aim of this study was to identify novel glycome quantitative trait loci (glyQTLs), prioritize novel candidate genes, and reconstruct tissue-specific gene networks that regulate plasma protein glycosylation. For this, we performed the largest genome-wide association meta-analysis (GWAMA) of total plasma N-glycome using data from seven studies (N = 10,764). For replicated glyQTLs, we prioritized candidate genes using a broad spectrum of methods and explored how these genes are connected in a functional tissue-specific network that regulates protein glycosylation. The second aim of this study was to identify

potential glycan biomarkers for disease. We performed a phenome-wide association study (PheWAS) in conjunction with colocalization analysis to investigate the pleiotropic effects of glyQTLs on complex diseases. Next, we correlated glycan polygenic scores in 450,000 UK Biobank samples with the disease's endpoints. Finally, we conducted a bidirectional Mendelian randomization study to identify potential causal effects between glycans and disease. This strategy not only resulted in the discovery of new candidate genes but also suggested how some of these genes might regulate glycosylation enzymes and how they are linked to the aberrant glycosylation observed in disease.

## Results

### Single- and multi-trait GWASs for 138 N-glycome traits

The levels of 36 N-glycan structures (Supplementary Data 1a) linked to various plasma glycoproteins were measured by ultra-high performance liquid chromatography in seven participating cohorts from six countries. The majority of 10,764 participants (94.5%) were of European ancestry. From the 36 directly measured N-glycans, we computed 81 derived N-glycome traits such as the total level of fucosylation, galactosylation, sialylation and others, reflecting pathways of N-glycan biosynthesis (Supplementary Data 1a). We conducted GWAS for each of these 117 N-glycome traits in each of the seven participating cohorts, assuming an additive model of the genetic effect. We then performed a fixed-effect discovery meta-analysis of the subcohorts, which comprised a total of 7,540 participants (Supplementary Data 2b). After meta-analysis, we took advantage of the correlation structure between 117 N-glycome traits and performed GWAS of 21 multivariate N-glycome traits defined based on their biochemical similarities (Supplementary Data 1b).

Our analyses identified and replicated a total of 40 loci (Fig. 1a, Supplementary Data 3a, b) that were significantly associated with at least one of 117 N-glycome traits and 21 multivariate N-glycome traits. The association of 25 loci with total plasma N-glycome was shown and replicated for the first time (Table 1), while the association of 15 loci confirms previous findings[36,37].

We performed an approximate conditional and joint analysis implemented in GCTA-COJO[44] to identify conditionally independent association signals in the replicated loci on discovery GWAMA. We found evidence of multiple SNPs contributing independently to glycan level variation for nine loci (Supplementary Data 4): two sentinel associations were observed in the loci containing fucosyltransferases *FUT8* and *FUT6*, sialyltransferases *ST6GAL1* and *ST3GAL4*, galactosyltransferase *B4GALT1*, glycuronyltranferase *B3GAT1*, and the acetylglucosaminyltransferase *MGAT5*. Beyond glycosyltransferase loci, the locus spanning the human leukocyte antigen (*HLA*) and the locus containing *HPR* gene showed secondary associations.

### SNP-based heritability and whole genome polygenic scores for 117 N-glycome traits

For 117 N-glycome traits we estimated SNP-based heritability using LD Score regression[45]. For 68 N-glycome traits SNP-based heritability was above zero at nominal $P \leq 0.05$, varying from 10.2% to 33.4% ($19.8 \pm 10.3\%$) (Supplementary Data 5a), which is on average 2.5x lower than the narrow-sense heritability of 37 N-glycome traits, estimated in a twins-based study – $50.6 \pm 14.0\%$[46].

For each of the 117 N-glycome traits, we created polygenic score (PGS) models based on the GWAMA of European-ancestry participants (N = 10,172) using the SBayesR method[47]. We tested the out-of-sample prediction accuracy of these models in the CEDAR dataset (N = 187 participants of European ancestry). For 79 N-glycome traits in CEDAR samples, PGS models explained from 2.4% to 20.0% of the trait variance (FDR < 5%), allowing for the calculation of glycan polygenic scores in large-scale cohorts of European descent (e.g., UKBiobank). For the remaining 38 N-glycome traits the explained variance did not deviate significantly from zero (FDR > 5%). The out-of-sample prediction accuracy correlated significantly with the SNP-based heritability

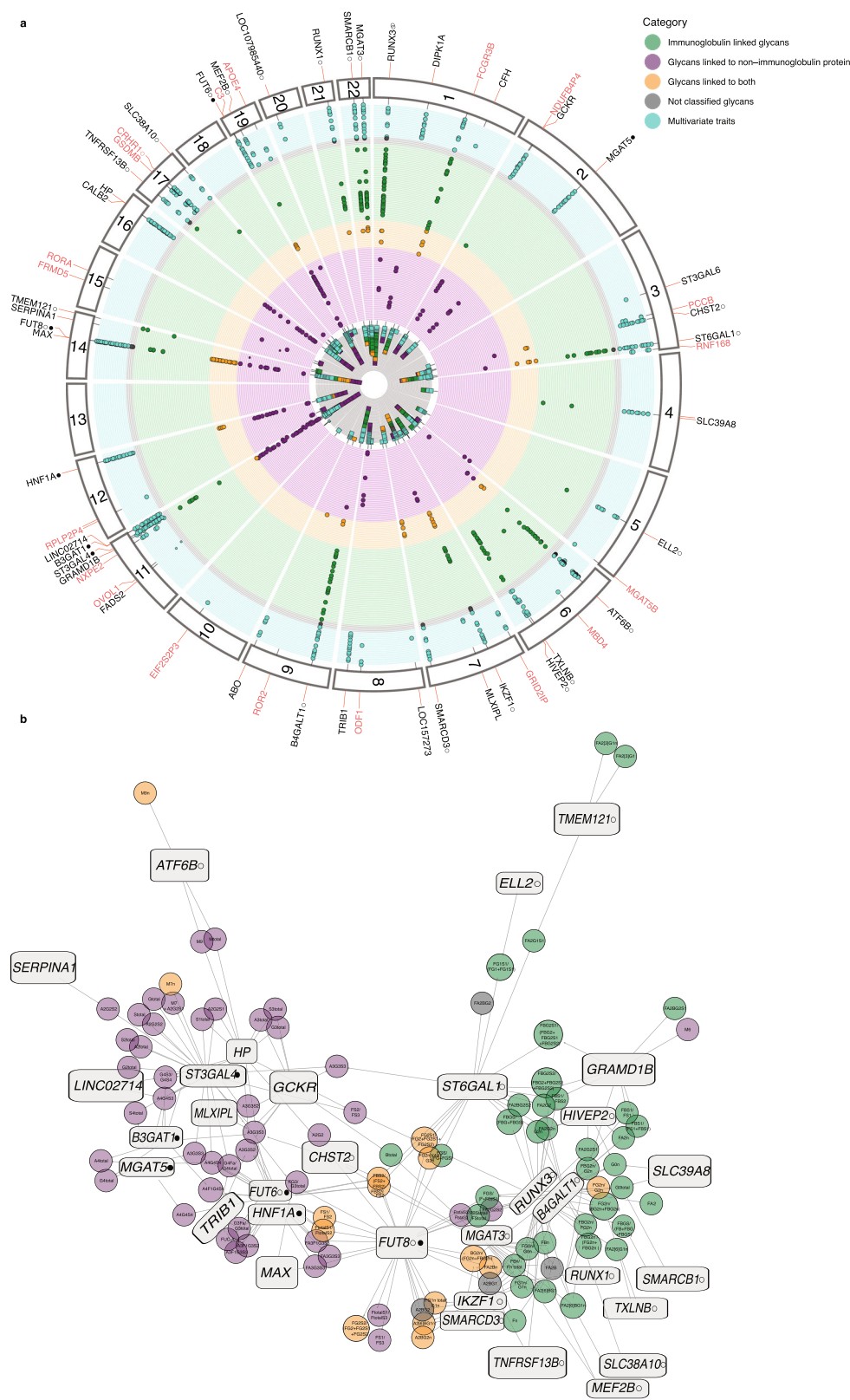

$(R = 0.48, P = 4.05 \times 10^{-8})$. The implementation of SBayesR models is detailed in Supplementary Data 6.

## Prioritization of causal genes for protein N-glycosylation

Identification of genes, rather than genetic loci, can help to find novel protein glycosylation regulators and suggest targets for intervention in glycome-related diseases. We prioritized candidate genes using a consensus approach integrating eight lines of evidence - based on a literature search of genes encoding known enzymes and regulators of N-glycan biosynthesis; genes causing congenital disorders of glycosylation; colocalization of glyQTLs with eQTLs and blood plasma pQTLs; annotation of putative causal variants affecting protein structure; enrichment of gene sets and tissue-specific expression; and prioritization of the nearest gene (see Methods)[48].

**Fig. 1 | Discovered loci. a** Associations of 59 loci with 138 glycomic traits labeled by the prioritized candidate or nearest gene names. Each dot corresponds to a trait-associated locus. Loci names marked black: loci are discovered and replicated in this work; red: discovered, but not replicated in this work. In total, 117 univariate traits were analyzed (Supplementary Data 1a), but for two of them, no genome-wide significant associations were found. Univariate traits are grouped into 4 categories: glycans mostly linked to immunoglobulins (green circles), glycans mostly linked to non-immunoglobulin proteins (purple circles), glycans linked to both types of proteins (orange circles), not classified glycans (gray circles). The details of glycan classification are described in the Supplementary Note. Also, the results from analysis of 21 multivariate traits (turquoise circles) are presented. The multivariate

traits were defined based on biochemical similarities between 36 directly measured total plasma N-glycan traits (Supplementary Data 1b). Thus, there are a total of 136 circles representing both univariate and multivariate N-glycan traits. **b** A network view of associations between loci and glycan traits. Rectangular nodes represent genetic loci labeled with the names of the prioritized candidate or nearest genes; circle nodes represent glycan traits. Lines represent significant genetic associations between locus and specific glycans. The colors of circle nodes are consistent with those in (**a**). Symbol ○ next to the candidate gene indicates that the locus was previously discovered in immunoglobulin G N-glycome GWASs; Symbol ● – the locus was previously discovered in transferrin N-glycome GWAS, as reviewed in ref. 25.

We prioritized candidate genes in 31 of the 40 glyQTLs (Supplementary Data 7a) by selecting the gene with the highest unweighted sum of evidence across all eight predictors[48], provided a gene was supported by at least two predictors. The prioritized genes may regulate the protein N-glycosylation through several known general mechanisms: biosynthesis of N-glycans, abundance of N-glycoproteins in the blood, regulation of transcription in lymphoid and gastrointestinal tissues, and ion homeostasis in the endoplasmic reticulum and Golgi apparatus.

Among the 31 prioritized genes (Fig. 2b), we identified nine genes encoding glycosyltransferases (*MGAT5, ST6GAL1, B4GALT1, ABO, ST3GAL4, B3GAT1, FUT8, FUT6, MGAT3*); mutations in three are known to lead to congenital disorders of glycosylation (*B4GALT1, FUT8, SLC39A8*) and four genes have strong experimental support for being regulators of N-glycan biosynthesis genes (*HNF1A, IKZF1, RUNX3, SLC39A8*)[25]. The SMR/HEIDI approach indicated that total plasma N-glycosylation–associated variants in two loci possibly had pleiotropic effects on plasma levels of two blood proteins (HPT, CAFH) (Supplementary Data 7a, f) and transcription of 10 genes in different tissues (Supplementary Data 7a, e). In 12 genes, associated variants were either coding or were in strong LD with the variants coding for potentially deleterious amino acid changes, and in 5 genes - pathogenic amino acid changes. The DEPICT gene prioritization tool[31] provided evidence of prioritization for 19 genes in 18 loci at FDR < 0.2 (Supplementary Data 7g). Additionally, 13 glyQTLs were found to colocalize with a trans-pQTL (see Supplementary Fig. 1); however, we did not include the trans-pQTL as a predictor in gene prioritization. Since colocalization between glyQTLs and pQTLs may reflect an effect of N-glycosylation on protein levels or technical artifacts of protein measurement, such as altered SomaLogic binding affinity, we conducted bidirectional MR between variations in blood protein levels (e.g., CFH, HP) and their N-glycan abundances (Supplementary Data 16b). The results showed a robust causal effect of CFH level on PGP18 (M9) and a causal effect of HP level on nine N-glycan traits: high-mannose (M9 and Mtotal), tri-antennary sialylated galactosylated N-glycans (A3G3S2, A3G3S3, FS2/FS3, S3total, G3total, A3total) and ratio between tri-sialyalted and tetrasialylated tetragalactosylateed N-glycans (G4S3/G4S4).

In the following discussion, we focus on thirteen novel candidate genes that were not yet confidently identified before in GWASs of human protein N-glycosylation; for information on those earlier studies, we refer the reader to previous works and published reviews[25,26,36,37,40,41]. We prioritized four genes associated with lipid metabolism regulation - *GCKR, FADS2, TRIB1,* and *GRAMD1B* which, to our knowledge, is the first time protein N-glycosylation has been linked to genes involved in lipid metabolism and its regulation; four genes encoding N-glycoproteins having anti-inflammatory function - *HP, HPR, SERPINA1* and *CFH*; the gene *SCL39A8* encoding a zinc transporter; three genes encoding transcription factors - *MAX, NFKB1, MYRF*; and a glycosyltransferase gene *ABO*, which determines an individual's ABO blood type.

The Supplementary Note provides an in-depth account of the details of thirteen newly prioritized genes. The other genes that have

been previously prioritized elsewhere are described in Timoshchuk et al. [25].

## Tissue-specific regulation of plasma protein N-glycosylation in liver and lymphoid tissue

Lymphoid tissue, specifically plasma cells that produce antibodies, and liver, specifically hepatocytes, contribute the majority of glycoproteins present in human blood[2,25] and are thus the primary drivers of N-glycosylation of plasma proteome. However, the N-glycosylation machinery in these two cell types varies, leading to distinct spectra of glycans attached to proteins produced in these two tissues[2,46]. Many glycosylation-associated genes prioritized in this study are expressed in plasma cells, or hepatocytes, or both (Fig. 2a).

To gain a deeper understanding of the tissue-specific regulation of glyco-genes, we constructed a gene network for N-glycosylation regulation. This network comprised 32 loci that were replicated in the univariate association analysis, and 117 N-glycome traits as vertexes, with significant associations between them represented as edges (Fig. 1b). The resulting network revealed two major subnetworks, wherein candidate genes and glycan traits were clustered. The first subnetwork was primarily associated with blood plasma N-glycans typically produced in the liver, and included 13 loci (*ATF6B, B3GAT1, CHST2, FUT6, HNF1A, HP, LINC02714, MAX, MGAT5, MLXIPL, SERPINA1, ST3GAL4, TRIB1*). The second subnetwork was related to blood plasma N-glycans typically attached to immunoglobulins and consisted of 14 loci (*B4GALT1, ELL2, HIVEP2, IKZF1, MEF2B, MGAT3, RUNX1, SLC38A10, SLC39A8, SMARCB1, SMARCD3, TNFRSF13B, TMEM121, TXLNB*). According to the classification of Clerc et al.[2], genetic variation in five loci (containing *FUT8, GCKR, GRAMD1B, RUNX3,* and *ST6GAL1*) had an impact on plasma N-glycans attached to both immunoglobulins and liver-secreted proteins. Most of these five loci exhibited strong bias towards N-glycans known to be preferentially expressed on proteins produced in one of the tissues, i.e., *GRAMD1B* was associated with 8 N-glycans, of which 7 were typical for liver proteins; *GCKR* – with 9, of which only one was typical for immunoglobulins; *RUNX3* and *ST6GAL1* were preferentially associated with N-glycans typically attached to immunoglobulins (32/35 and 43/44 glycans, respectively). It should be noted that the classification of Clerc et al.[2] was compiled based on a large body of literature data, and we cannot exclude occasional misclassification.

To gain insight into the spectrum of glycans that were associated with the 8 loci that were replicated in multivariate association analysis, we considered significant at ($p < 0.01/36$) association of the partial regression coefficients in the multivariate analysis of trait set "N-glycosylation" (36 traits) (Supplementary Data 8c). In this analysis, *DIPK1A, FADS2,* and *CALB2* loci are associated with N-glycans typical for liver-secreted proteins; *LOC107985440* – with N-glycans typically observed on immunoglobulins IgG. Results for *ST3GAL6* and *LOC157273* were inconclusive, although the former was associated with the multivariate trait "high branching N-glycans"; such glycans are typical for liver-secreted proteins.

Three loci showed clear effect on N-glycans found on both liver-secreted glycoproteins and immunoglobulins: *FUT8* (significant effect

**Table 1 | Twenty-five novel loci associated with total plasma N-glycosylation discovered and replicated in this study**

| SNP info | | | | Discovery | | | | | | |
|---|---|---|---|---|---|---|---|---|---|---|
| SNP | CHR:POS | EA/RA | Gene | N | EAF | BETA (SE) | P | Top trait | Type of association |
| rs12726286 | 1:93334379 | C/T | DIPK1A | 7540 | 73.52% | 0.022 (0.002) | 2.24E-21 | sialylation of antennary branches | multivariate |
| rs1329427 | 1:196704559 | C/T | CFH | 7540 | 57.63% | 0.016 (0.003) | 5.61E-10 | N-glycosylation | multivariate |
| rs1260326 | 2:27730940 | C/T | GCKR | 7081 | 57.51% | -0.168 (0.018) | 9.65E-22 | G3total | univariate |
| rs2470750 | 3:98690592 | A/T | ST3GAL6 | 6790 | 59.04% | 0.012 (0.002) | 7.23E-10 | high branching glycans | multivariate |
| rs3774964 | 4:103519487 | A/G | SLC39A8 | 7164 | 63.69% | 0.109 (0.018) | 1.49E-09 | G0n | univariate |
| rs7705720 | 5:95280033 | C/T | ELL2 ○ | 7540 | 20.03% | -0.133 (0.021) | 1.32E-10 | FG1S1/(FG1 + FG1S1) | univariate |
| rs4543384 | 6:139629524 | C/T | TXLNB ○ | 6790 | 55.74% | 0.143 (0.018) | 9.67E-16 | FA2BG1n | univariate |
| rs7758383 | 6:143169723 | A/G | HIVEP2 ○ | 7540 | 49.87% | 0.153 (0.017) | 3.55E-20 | FA2G2n | univariate |
| rs34166762 | 7:73018524 | C/T | MLXIPL | 7540 | 27.58% | -0.121 (0.019) | 1.09E-10 | A3G3S2 | univariate |
| rs7781265 | 7:150950940 | A/G | SMARCD3 ○ | 7540 | 10.99% | 0.223 (0.027) | 4.75E-16 | M64 | univariate |
| rs4841133 | 8:9183664 | A/G | LOC157273 | 7540 | 8.37% | 0.013 (0.002) | 6.85E-10 | sialylation of antennary branches | multivariate |
| rs28601761 | 8:126500031 | C/G | TRIB1 | 6790 | 57.56% | -0.143 (0.018) | 7.10E-15 | G3Fa/G3total | univariate |
| rs58218 | 9:136145471 | A/G | ABO | 7540 | 65.12% | 0.023 (0.003) | 2.00E-17 | N-glycosylation | multivariate |
| rs174528 | 11:61543499 | C/T | FADS2 | 7540 | 35.83% | 0.013 (0.002) | 1.50E-09 | sialylation of antennary branches | multivariate |
| rs36020612 | 11:123344435 | C/T | GRAMD1B | 7319 | 79.96% | -0.169 (0.022) | 1.66E-14 | FBG2S1/ (FBG2 + FBG2S1 + FBG2S2) | univariate |
| rs11223982 | 11:134612702 | A/G | LINC02714 | 6280 | 12.47% | 0.256 (0.029) | 3.45E-19 | A4G4S3 | univariate |
| rs7161378 | 14:65450780 | C/T | MAX | 7540 | 24.74% | -0.180 (0.019) | 2.05E-20 | FG3/G3total | univariate |
| rs28929474 | 14:94844947 | C/T | SERPINA1 | 5135 | 97.71% | 0.437 (0.072) | 1.01E-09 | A2G2S2 | univariate |
| rs8046823 | 16:71400131 | A/G | CALB2 | 7540 | 51.53% | 0.021 (0.003) | 1.46E-16 | galactosylation of antennary branches | multivariate |
| rs217184 | 16:72105965 | C/T | HP | 7540 | 19.90% | 0.211 (0.021) | 5.44E-23 | S3total | univariate |
| rs4500785 | 17:16848565 | C/G | TNFRSF13B ○ | 7540 | 88.63% | -0.180 (0.027) | 1.33E-11 | FA2BG1 | univariate |
| rs2659007 | 17:79217478 | A/G | SLC38A10 ○ | 7540 | 46.44% | 0.108 (0.017) | 4.07E-10 | FA2BG1n | univariate |
| rs11669860 | 19:19277296 | A/G | MEF2B ○ | 7540 | 55.43% | -0.113 (0.017) | 7.14E-11 | FA2BG1n | univariate |
| rs2618588 | 20:17832658 | C/T | LOC107985440 ○ | 7540 | 39.69% | 0.011 (0.002) | 2.02E-10 | core-fucosylation | multivariate |
| rs2834847 | 21:36588180 | A/C | RUNX1 ○ | 7540 | 77.22% | 0.160 (0.020) | 2.58E-15 | FBn | univariate |

Full results of discovery and replication are provided in Supplementary Data 3a, b. CHR:POS—chromosome and position of SNP according to GRCh37 human genome build; EA/RA—effective and reference allele; Gene—prioritized or nearest gene for a locus (Supplementary Data 7a); N—sample size; EAF—effective allele frequency; BETA (SE)—effect (in SD units) and standard error of effect; P—P-value of the effect estimate (two-sided Wald test with one degree of freedom. Genome-wide significance threshold was adjusted for multiple tests for univariate type of association, MANOVA test with k degrees of freedom for multivariate type of association, where k is a number of glycans in a multivariate trait); Top trait—glycan trait with the strongest association (the lowest P); Type of association—univariate or multivariate. Description of glycan traits is provided in Supplementary Data 1. Symbol ○ — the locus was previously discovered in immunoglobulin G N-glycome GWASs, as reviewed in ref. 25.

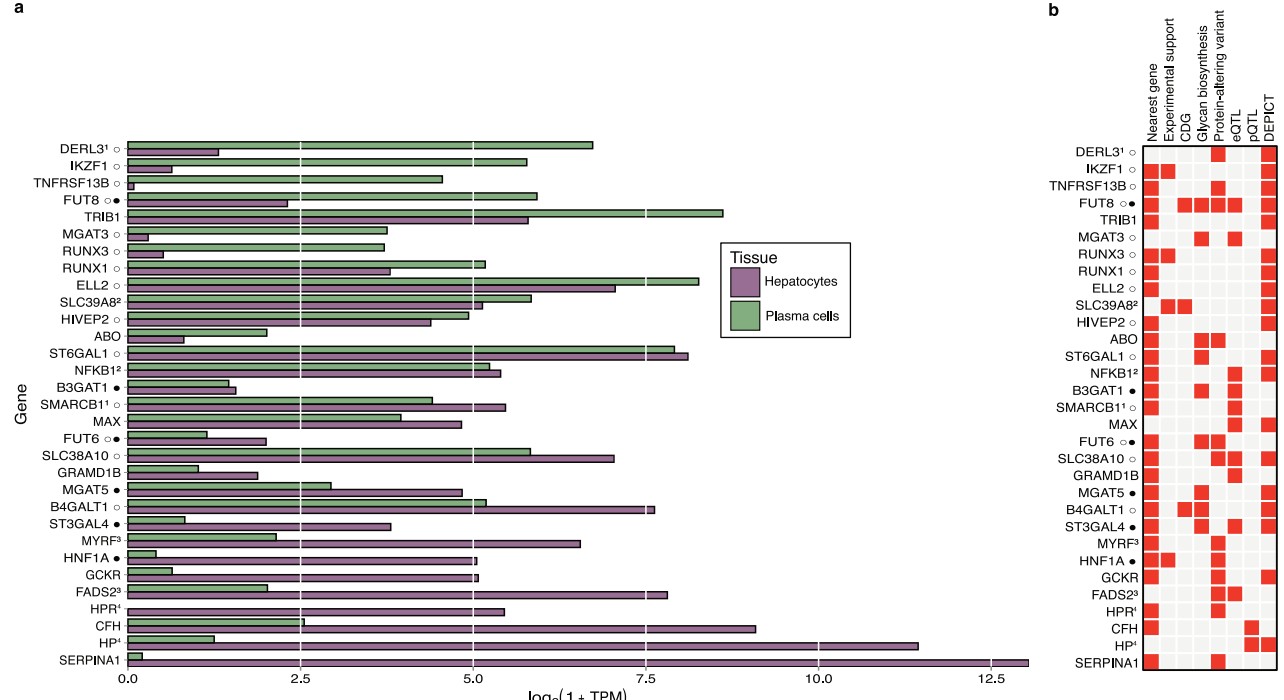

**Fig. 2 | Candidate genes. a** Gene expression of the candidate genes in two relevant cell types - hepatocytes and plasma cells. Expression levels are represented as the median logarithm of transcripts per million. The data for hepatocytes ($N = 513$) and plasma cells ($N = 53$) samples were obtained from the ARCHS4 portal[101]. **b** Predictors indicating the 32 candidate genes. Gene order corresponds to (**a**). The identical superscripts denote candidate genes inside one locus. Full details of the gene prioritization are presented in Supplementary Data 7a. Symbol ○ next to the candidate gene indicates that the locus was previously discovered in immunoglobulin G N-Glycome GWASs; Symbol ● − the locus was previously discovered in transferrin N-Glycome GWAS, as reviewed in ref. 25.

on 9 N-glycans typically attached to liver-secreted proteins and 3 typically attached to immunoglobulins); *CFH* (2 and 2, respectively) and *ABO* (also 2 and 2).

TF and IgG are two proteins secreted by hepatocytes and plasma cells, respectively, and GWASs of their N-glycosylation shed light on the genetic control of protein N-glycosylation in the corresponding tissues[42]. To gain further insights into the mechanism of association and to support the tissue-specificity of the loci, we conducted a colocalization analysis of total plasma, IgG, and TF glyQTLs using the SMR-θ method[49]. The analysis was restricted to the loci that were previously implicated in TF[42] N-glycome or IgG[40] N-glycome GWASEs, and reached genome-wide significance in univariate association analysis in this study. Excluding *HLA*, this selection resulted in 21 loci, of which 15 were significant in previous IgG N-glycome GWAS only, four were only significant in the TF N-glycome GWAS, and two (*FUT6* and *FUT8*) were significant in both (Supplementary Data 8b)[25]. For specific locus, we colocalized signals of genetic association for traits that have reached genome-wide significance in that locus.

The results of colocalization analysis are presented in Supplementary Fig. 4. If regional genetic associations of a plasma N-glycome trait colocalized ($|\theta| > 0.7$) with genetic associations of an IgG N-glycome trait, we considered this as evidence that the locus is expressing its effect on plasma N-glycome through its effect on IgG N-glycosylation, acting in antibody-producing cells. Similarly, colocalization with genetic association signal for TF N-glycome was taken as an indication that the locus may exhibit its action via the effect of TF N-glycome, acting in the liver. The analysis suggested that the *ELL2, TXLNB, HIVEP2, IKZF1, SMARCD3, TMEM121, SLC38A10, MEF2B, ATF6B, RUNX1, RUNX3, SMARCB1, MGAT3, ST6GAL1, B4GALT1* loci regulate N-glycosylation of IgG while *MGAT5, ST3GAL4, B3GAT1, HNF1A* loci regulate N-glycosylation of TF. The *FUT8 and FUT6* act as regulators of both glycoproteins. An interesting case of pleiotropy was observed in the *FUT8* locus. The colocalization signal in *FUT8* split into two distinct clusters

(Supplementary Fig. 4, page 120), one of which was dominated by N-glycans predominantly presented on proteins produced in the liver, while the other was almost exclusively presented by these on immunoglobulins. To support the hypothesis of two distinct tissue-specific genetic mechanisms in the locus, we combined traits from the two clusters into single traits using MANOVA approach[28] and performed a colocalization analysis between the two constructed linear combinations using the SMR-θ method[49] and R Coloc package[50]. We found strong evidence *against* colocalization of N-glycans presented on liver-specific proteins and these on immunoglobulins in this genomic region (PP.H3 = 100%, where PP.H3 is the posterior probability that there are two distinct causal variants contributing to trait variation, $\theta = 4 \times 10^{-6}$). This supports the hypothesis that genetic regulation of *FUT8* in the liver and antibody-producing cells follows two distinct mechanisms, as previously suggested by Landini and colleagues[42].

Thus, the colocalization analysis confirms different tissue-specific mechanisms of genetic regulation for each locus. With the exception of the *ATF6B* and *FUT6* loci, the results from colocalization are largely consistent with the classification based on gene-N-glycan association network.

Our findings demonstrate that the genetic regulation of protein N-glycosylation is highly tissue-specific. Even glycosyltransferases such as *FUT8, FUT6, MGAT3, ST6GAL1*, and *B4GALT1*, being expressed in antibody-producing cells and hepatocytes (Fig. 2a) and known to participate in the N-glycan biosynthesis in both tissues, display pronounced tissue-specific genetic regulation that is not shared between different tissues.

## PheWAS highlights loci associated with an extensive number of diseases and quantitative traits

In this study, we examined the pleiotropic effects of 40 replicated glyQTLs on over a thousand diseases and quantitative traits endpoints (as listed in Supplementary Data 9a) using the SMR/HEIDI method. Our

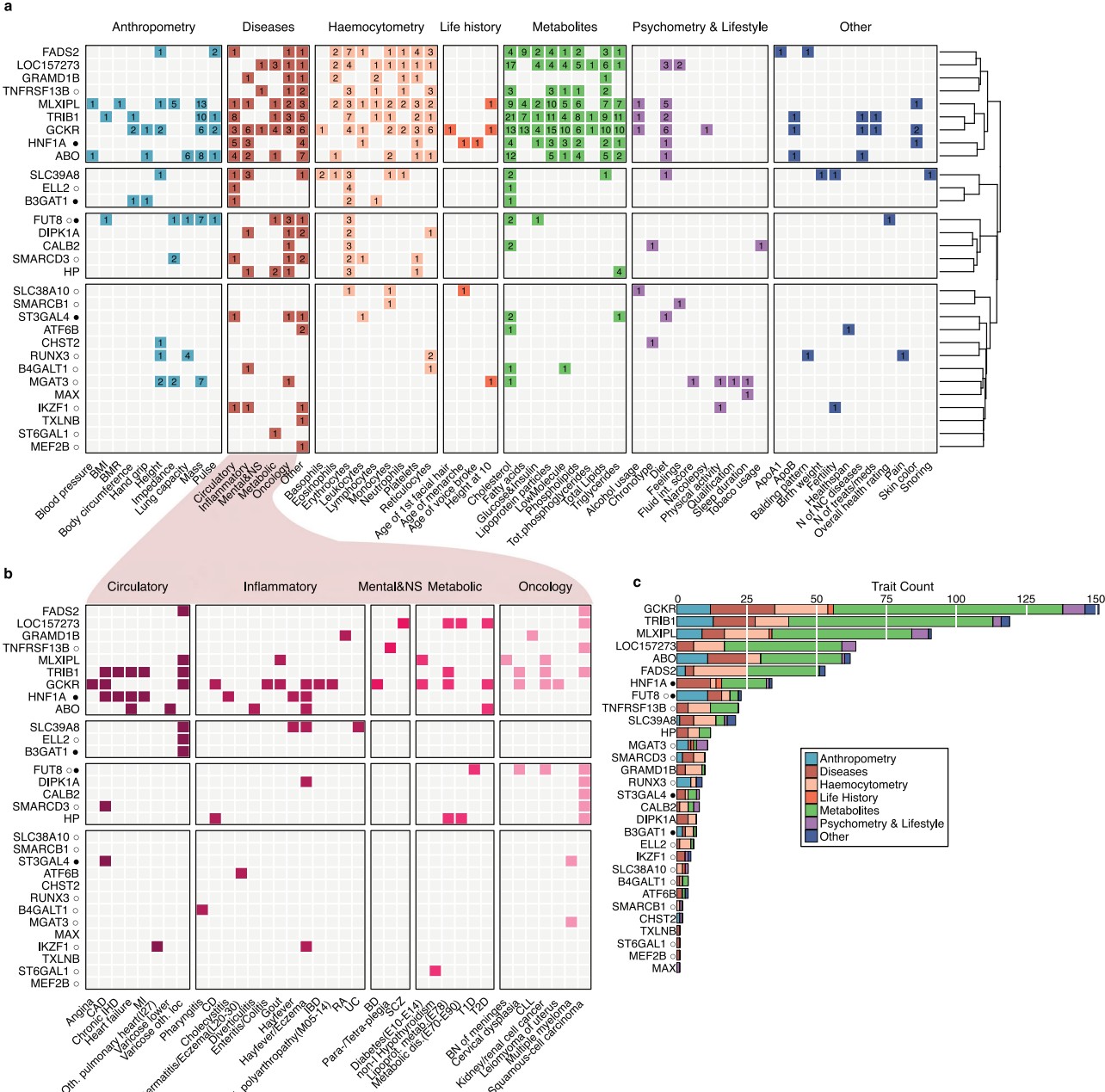

**Fig. 3 | Phenome-wide colocalization results for significant and replicated loci.**
**a** Heatmap of traits with pleiotropic associations ($P_{adjSMR} < 0.05$; $P_{HEIDI} \geq 0.05$, $P_{HEIDI}$ is p-value for HEIDI test; $P_{adjSMR}$ is two-sided z-test with Benjamini-Hochberg adjusted p-value for the estimated SMR effect.). The row order is determined by clustering on the set of significantly colocalized traits (Jaccard similarity index and Ward's linkage). For the sake of readability, similar traits are grouped into broader categories, e.g., such traits as fat-free mass of left leg, trunk fat mass are combined under the name "Body mass". The numbers in the cells represent the number of pleiotropic associations, grouped under a common name, confirmed for a given locus. Color does not carry semantic load. **b** Heatmap providing details of pleiotropic associations with diseases from (**a**). Ordering is the same as in (**a**), but loci with no colocalized diseases are omitted. Color does not carry semantic load. **c** Count of traits for each locus with pleiotropic associations. Symbol ○ next to the candidate gene indicates that the locus was previously discovered in immunoglobulin G N-Glycome GWASs; Symbol ● − the locus was previously discovered in transferrin N-Glycome GWAS, as reviewed in ref. 25. BMI body mass index, BN benign neoplasm, CLL chronic leukocytic leukemia, T1D type 1 diabetes, T2D type 2 diabetes, BD bipolar disorder, BP blood pressure, MI myocardial infarction, CAD coronary artery's disease, CD Crohn disease, IBD Inflammatory bowel disease, IHD Ischemic heart disease, NC non-cancer, non-I non-iodine dependent, RA rheumatoid arthritis, SCZ schizophrenia, UC ulcerative colitis.

analysis revealed a total of 1,214 significant associations, of which 781 demonstrated a non-rejection of the pleiotropy hypothesis by the HEIDI test. The identified pleiotropic associations encompassed a wide range of phenotypes, including type 2 and type 1 diabetes, blood glucose levels, coronary artery disease, cholesterol levels, bipolar disorder, schizophrenia, gout, various oncological diseases, metabolomic and anthropometric traits, lifestyle and diet-related traits,

general life history and overall health status, among others (as depicted in Fig. 3a, b and Supplementary Data 9b). Additionally, *TRIB1* and *GCKR* showed colocalization with metabolic dysfunction-associated steatotic liver disease ($P_{SMR} = 4.58 \times 10^{-8}$, $P_{HEIDI} = 0.03$ (possibly shared); $P_{SMR} = 3.26 \times 10^{-8}$, $P_{HEIDI} = 0.73$ (likely shared)).

Hierarchical clustering based on sets of colocalized traits allowed us to differentiate four distinct groups of loci (Fig. 3a, right panel). The

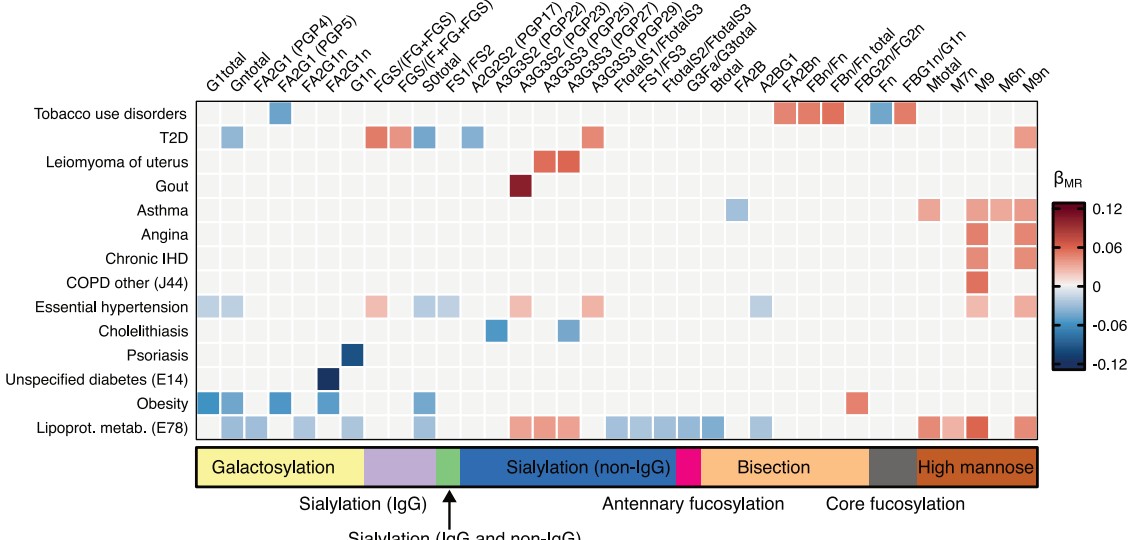

**Fig. 4 | Significant associations between PGS for plasma N-glycosylation traits and ICD-10 diseases.** Heatmap of associations between PGS for plasma N-glycosylation traits and ICD-10 diseases that were statistically significant at the designated significance threshold of $1.07 \times 10^{-5}$. Every column represents an N-glycosylation trait, every row – an ICD-10 disease. The color of the cell represents the effects estimated in the logistic regression of the disease phenotype incidence on the PGS for each glycan trait – blue hues represent negative effects, red hues – positive, and the intensity of the color represents the absolute value of the effect (larger values are shown in darker hues). Gray cells represent the non-significant associations. The bands under the heatmap depict the groups of N-glycosylation traits that are related to certain structural features of the N-glycans. T1D – type 1 diabetes.

first cluster (Fig. 3a, topmost cluster) was characterized by a high number of metabolic colocalizations and encompassed eight of the ten most pleiotropic loci (Fig. 3c). It was also found to be colocalized with a diverse range of other phenotypes, including disease and anthropometric traits. Of particular note, this cluster comprised loci with prioritized lipid metabolism genes, namely, *GCKR, TRIB1, FADS2* and previously known *HNF1A*. Furthermore, it encompassed the locus containing the *MLXIPL* gene, a transcriptional factor that induces liver glycolysis and lipogenesis, as well as *ABO*, which is known to be associated with stroke[51], metabolic dysfunction-associated steatotic liver disease and levels of lipids[52,53].

## Association between PGS for plasma N-glycosylation traits and ICD-10 diseases

We analyzed associations between polygenic scores (PGS) for the 117 plasma N-glycosylation traits and 167 diseases classified according to International Classification of Diseases (ICD)-10 in individuals of European ancestry from the UK Biobank cohort ($N = 374{,}303$) (Supplementary Note). The analysis revealed 14 diseases associated with PGS for at least one plasma N-glycome trait and PGS for 35 plasma N-glycome traits associated with at least one disease at the designated significance threshold of $P < 1.07 \times 10^{-5}$ (Fig. 4, Supplementary Data 10). Full results and overview of the analysis are provided in the Supplementary Note.

Notably, we observed positive associations between PGS for traits, related to the increased levels of high-mannose glycans, especially to those of containing nine mannose residuals (M9), with cardiovascular disease phenotypes such as essential hypertension, ischemic heart disease, angina, hyperlipidaemias, as well as type 1 diabetes and asthma (Fig. 4).

We found negative associations for the PGS for S0 total (percentage of neutral N-glycan structures, i.e. N-glycans without sialic acid, in total plasma N-glycans) with primary hypertension, lipidaemias, obesity and non-insulin dependent diabetes (Fig. 4). Moreover, PGS for the N-glycome traits describing the abundances of galactosylated structures were negatively associated with obesity, lipoprotein metabolism disorders, primary hypertension and type 2 diabetes (Fig. 4).

## Bidirectional Mendelian Randomization analysis of causal effects between plasma N-glycosylation traits and ICD-10 diseases

For the statistically significantly associated pairs of disease-plasma N-glycome traits in UK Biobank we conducted bidirectional two-sample Mendelian randomization (MR) analysis to investigate the direction of effects.

Using the disease phenotype as exposure, we conducted a two-sample MR analysis and revealed statistically significant positive causal effect of disorders of lipoprotein metabolism on M9 (N-glycan with nine mannose residuals) and on Mtotal (the percentage of high-mannose structures in total plasma glycans) (Table 2, Supplementary Data 11b, Supplementary Figs. 7a–d, 8a–d). The direction of the causal effect corresponded to the signs of the beta of association between disorders of lipoprotein metabolism and other lipidaemias (E78) and PGS for M9 and Mtotal. Sensitivity analyses, including a two-sample MR after removal of pleiotropic IVs, as well confirmed the observed causal effects of lipidaemias (E78) on M9 and Mtotal (Supplementary Figs. 9a–d, 10a–d; Supplementary Data 12a–c, 14a–c, Supplementary Note).

Using plasma N-glycome trait as exposure we performed MR analysis in the opposite direction, discovering a statistically significant positive causal effect of M6n, percentage of M6 in total neutral plasma glycans on asthma and a positive effect of M9n, percentage of M9 in total neutral plasma glycans, on disorders of lipoprotein metabolism and other lipidaemias (Table 3, Supplementary Data 11a, Supplementary Figs. 5a–d; 6a–c). In both cases, the direction of the effect was concordant with the direction of association between corresponding pairs of disease and PGS. Sensitivity analyses also confirmed the observed causal effect (Supplementary Data 12a–c). Since the number of available IVs for both M6n and M9n was not sufficient for the analysis of pleiotropy among the IVs using MR-PRESSO[54], as an additional sensitivity analysis for the effects of M6n on asthma and M9n on lipidaemia we performed colocalization (SMR-HEIDI) analysis for the loci tagged by the genetic variants used as IVs in these cases (Supplementary Data 13). SMR-HEIDI analysis provided evidence for one shared causal variant (rs144126567, $p_{SMR} = 0.003$; $p_{HEIDI} = 0.30$) for M6n and asthma but did not find any proof for existence of shared

**Table 2 | Causal associations between plasma N-glycosylation traits and ICD-10 diseases discovered in this study**

| Exposure | Outcome | Analysis type | MR method | Number of IVs | Causal BETA (SE) | P |
|---|---|---|---|---|---|---|
| E78 - Disorders of lipoprotein metabolism and other lipidaemias | PGP18 - The percentage of M9 | Primary | Inverse variance weighted | 27 | 4.09 (1.1772) | 5.16E-04 |
| | | Sensitivity | Inverse variance weighted | 19 | 3.03 (0.82) | 6.19E-05 |
| E78 - Disorders of lipoprotein metabolism and other lipidaemias | PGP107 - The percentage of high-mannose structures in total plasma glycans | Primary | Weighted median | 27 | 3.01 (0.8756) | 5.87E-04 |
| | | Sensitivity | Weighted median | 23 | 3.13 (0.87) | 3.17E-04 |

Full results of the analysis are provided in Supplementary Data 11a, b. Exposure – upstream trait in the analysis (cause); Outcome – downstream trait in the analysis; IV- instrumental variable (independent SNPs associated with upstream trait); Analysis type – primary (exploratory) or sensitivity (after removal of pleiotropic IVs). See Methods for details, Number of IVs – number of IVs used for the analysis; Causal BETA (SE) – causal effect of the exposure on the outcome (in log OR units of the disease unit per SD units of glycan trait if glycan trait was an exposure) with its standard error; P – two-sided Bonferroni-corrected p-value of the MR effect estimate.

**Table 3 | Causal associations between ICD-10 diseases and plasma N-glycosylation traits discovered in this study**

| Primary MR Analysis | | | | | | SMR-HEIDI (Sensitivity analysis) | | | | |
|---|---|---|---|---|---|---|---|---|---|---|
| Exposure | Outcome | MR method | Number of IVs | Causal BETA (SE) | P | Marker SNP (Top SNP*) | Beta SMR (SE) | P-value SMR** | P-value HEIDI | Conclusion |
| PGP64 - The percentage of M6 in total neutral plasma glycans (GPn) | J45 - Asthma | Weighted median | 3 | 0.012 (0.0036) | 9.78E-04 | rs144126567 | 0.0604 (0.021) | 0.016 | 0.2977 | Evidence for causal relationships confirmed by SMR-HEIDI at rs144126567 |
| | | | | | | rs614004 (rs617119) | 0.0186 (0.025) | 0.4525 | 0.182 | |
| | | | | | | rs440674 | 0.0627 (0.025) | 0.0222 | 0.0135 | |
| PGP69 - The percentage of M9 in total neutral plasma glycans (GPn) | E78 - Disorders of lipoprotein metabolism and other lipidaemias | Inverse variance weighted | 2 | 0.013 (0.0035) | 1.25E-04 | rs401775 | 0.0485 (0.020) | 0.0222 | 0.0418 | No evidence for causality by SMR-HEIDI at any loci |
| | | | | | | rs7529425 (rs140970775) | 0.0316 (0.020) | 0.1423 | 0.0526 | |

Full results of the analysis are provided in Supplementary Data 11a, b. Exposure – upstream trait in the analysis (cause); Outcome – downstream trait in the analysis; IV- instrumental variable (independent SNPs associated with upstream trait); Number of IVs – number of IVs used for the analysis; Causal BETA (SE) – causal effect of the exposure on the outcome (or in SD units of glycan trait per log OR unit of the disease phenotype) with its standard error; P – P-value of the MR analysis; Marker SNP - rsID of the SNP, which is IV; Beta (SMR) - beta MR of single IV and its standard error; P (SMR) - two-sided Bonferroni-corrected p-value of the MR effect estimate; P-value HEIDI - P-value of the HEIDI test.

genetic variants influencing M9n and lipidaemia (Supplementary Data 13). Therefore, we report an observed positive effect of M6n on asthma, while the presence of a positive effect of M9n on lipidaemia remains inconclusive. Full results of MR analysis are presented in Supplementary Data 11a, b.

As the result of PheWAS, we showed an overlap between genes associated with plasma protein N-glycosylation and those linked to liver diseases. To further investigate whether specific N-glycan traits may be causally associated with biomarkers of liver physiology, we conducted additional bidirectional Mendelian randomization (MR) analyses between N-glycans and five phenotypes that reflect distinct aspects of liver function: aspartate aminotransferase (AST), alanine aminotransferase (ALT), alkaline phosphatase (ALP), gamma-glutamyl transferase (GGT), and non-alcoholic fatty liver disease (NAFLD) (Supplementary Data 16a). Significant and robust MR associations were observed for AST and five N-glycans: PGP27 (A3G3S3, showing the strongest effect size; $\beta_{MR} = -0.06138$), PGP25 (also A3G3S3), PGP30 (A3F1G3S3), PGP92 (FUC-A, antennary fucosylation), and PGP113 (G4Fa/G4total, proportion of antennary fucosylated N-glycans among tetra galactosylated). We also observed a significant association between PGP108 (total level of bisected N-glycans) and GGT that passed sensitivity analysis; however, due to strong heterogeneity among genetic instruments ($P_{het} = 3.11 \times 10^{-109}$), this and other associations for ALP and GGT did not meet our criteria for robust MR results.

## Discussion

Here, we reported 40 quantitative trait loci (glyQTLs) discovered in the GWAS of 138 blood plasma N-glycome traits, resulting in a more than two-fold expansion of loci affecting N-glycosylation of blood plasma proteins. The integration of these findings with genetic information related to human diseases and other complex phenotypes allowed us to show for the first time that genes involved in liver function are linked to the human blood plasma protein N-glycosylation.

A subset of newly prioritized genes allows us to postulate a link between genetic regulation of metabolic and liver diseases and blood plasma protein N-glycosylation. Specifically, common genetic variation in the loci near *GCKR* and *TRIB1* is known to predispose to metabolic dysfunction-associated steatotic liver disease (MASLD)[55]. Moreover, genetic association signals for MASLD in these loci are colocalized with the corresponding glyQTLs. In the gene *SERPINA1*, rare Mendelian mutations lead to alpha-1 antitrypsin (AAT) deficiency, with liver disease as part of the phenotype. Common variation in this region associates with chronic elevation of alanine aminotransferase (cALT) levels[56], a proxy phenotype for MASLD.

We found positive associations between cardiovascular disease and PGS for high-mannose glycans, previously suggested as biomarkers for cardiovascular events in patients with diabetes[57], and negative associations between metabolic disorders and PGS for galactosylated glycans. MR analysis indicated a causal effect of M9 and Mtotal glycans on disorders of lipoprotein metabolism, and M6 glycan on asthma. Elevated high mannose glycans (e.g. M9, M6) may reflect disrupted glycan processing in the Golgi apparatus[58,59], a common feature in various pathological conditions, including cancer[60]. Alternatively, their increased plasma abundance could result from elevated levels of specific carrier proteins. For instance, apolipoprotein B-100, abundant in high-mannose structures (M9)[2], is closely linked to lipid disorders and cardiovascular disease[61–66], whereas elevated IgE antibodies, which prominently carry M6 glycans[2], play a role in allergic inflammation and asthma[67].

Although on the phenotypic level, the changes of total and liver secreted protein-specific N-glycosylation in liver disease are well-known[68–70], we demonstrate, for the first time, that specific genes are associated with both N-glycosylation and liver disease, offering a starting point for genetically-guided investigation of the functional mechanisms of this phenotypic association. We further showed (see

Supplementary Data 16a) that higher levels of the tri-sialylated N-glycan A3G3S3 and lower levels of antennary fucosylated glycans (A3F1G3S3, A4F1G4S4) are associated with lower aspartate aminotransferase (AST), a marker of hepatocyte integrity. These N-glycans are predominantly found on hepatocyte-secreted alpha-1-acid glycoprotein, alpha-1-antitrypsin, and beta-2-glycoprotein 1[25], which play roles in systemic inflammation, oxidative stress, and immune responses—processes linked to liver injury and repair[71]. Interestingly, A3G3S3 and antennary fucosylated N-glycans (A3F1G3S3 and A4F1G4S4) show opposite associations with AST levels, suggesting contrasting roles (as a part of N-glycoproteins) in liver physiology. Alternatively, the observed pattern may reflect a positive association between antennary fucosylation of hepatocyte-secreted proteins and hepatocellular injury. The latter is supported by recent findings of antennary fucosylation of haptoglobin being associated with systemic inflammation and various diseases[72]. However, we did not find significant robust causal associations for alanine aminotransferase (ALT), alkaline phosphatase (ALP), gamma-glutamyl transferase (GGT), or non-alcoholic fatty liver disease (NAFLD). This may result from limited statistical power due to the small number of genetic instrumental variables, or there might be a specific biological effect linking these glycans to AST levels, but not necessarily to other liver enzymes or NAFLD. While our data supports a genetic association between the A3G3S3 glycan structure and AST, further studies with enhanced statistical power are needed to clarify potential implications for liver health.

Somewhat superficially, we may reason that liver disease is characterized by hepatocyte injury and endoplasmic reticulum stress[73], which is strongly associated with changes in glycosylation. A less direct mechanism could be inflammation, which is a hallmark of liver disease, with proinflammatory cytokines shown to alter the substrate synthesis pathways as well as the expression of glycosyltransferases required for the biosynthesis of N-glycans[74]. Thus, changes in N-glycosylation observed in blood plasma may be at least partly explained by altered N-glycosylation of hepatocyte-secreted proteins. Consistent with this hypothesis, we demonstrate that common genetic variation in the three loci known to be associated with liver disease is associated with variation in the abundance of N-glycans, typically attached to the liver-secreted proteins.

Other notable, partly overlapping, subset of liver-expressed genes newly implicated in plasma protein N-glycosylation encodes anti-inflammatory proteins -- haptoglobin (HP, *HP*), complement factor H (CFAH, *CFH*), and alpha-1-antitrypsin (AAT, *SERPINA1*). The glyQTLs located at *HP* and *CFH* are colocalized with the corresponding cis-pQTLs. At least two mechanisms may be suggested to explain such colocalization. Genetic variants in these loci may affect the composition of blood N-glycosylation through changes in the abundance of glycans preferentially bond to HP and CFAH by regulating the level of these glycoproteins in the blood (the first hypothesis). Alternatively, the genetic variation may change glycosylation of these proteins, which, in turn, could change the affinity of the binding of the specific probes used by the SomaLogic assays (the second hypothesis). While we did not observe colocalization between *SERPINA1* glyQTL and the AAT pQTL, this may be a false negative, potentially explained by the low frequency of the *SERPINA1* lead variant (rs28929474). Nonetheless, the rs28929474 is associated with the level of glycoprotein acetyl, a mixture of N-glycoproteins, predominantly alpha-1-acid glycoprotein, haptoglobin, and alpha-1-antitrypsin[75]. We note that these two hypotheses are not mutually exclusive and may occur simultaneously. Bidirectional MR analyses indicated (Supplementary Data 16b) that variations in blood protein levels (e.g., CFH, HP) causally influence their N-glycan abundances, rather than N-glycans influencing protein levels. It should be noted that 15 genes in the corresponding glyQTLs, including the glycosyltransferase genes *ABO*, *ST3GAL4*, *MGAT3*, and *FUT8*, are colocalized with trans pQTLs, with *ABO* and *ST3GAL4* being known hotspots of trans-pQTL effects[76]. This is in line with the

hypothesis that technical variability in SOMALogic and Olink assays may be influenced by N-glycosylation effects on binding between proteins and probes[76]. However, our results do not conclusively prove that glycosylation alterations are the primary mechanism behind the trans-pQTL signals.

While variation at *HP* and *SERPINA1* loci is associated with changes in N-glycans typically attached to liver proteins, *CFH* locus appears to also affect N-glycans typically attached to immunoglobulins. While *CFH* does express at some level in plasma cells (Fig. 2a), we speculate that perhaps a more likely mechanism is that genetic variation in *CFH* affects its expression liver, and changes in N-glycans attached to immunoglobulins occur through a systemic mechanism, i.e., regulation of inflammation. Consistent with this hypothesis is the known association of the *CFH* locus with IgA nephropathy[77], as well as indications that in mice, CFH modulates splenic B cell development and limits autoantibody production[78].

Our results suggest that genetic regulation of plasma protein N-glycosylation predominantly occurs in lymphoid and liver tissue and exhibits strong tissue specificity. Integration of evidence from transcriptomics and N-glycomics suggests that molecular expression of genetic variation in the majority of glyQTLs is restricted to one tissue; the effects of this variation on N-glycans in blood plasma occur either through changes of N-glycosylation of the proteins secreted by the tissue, or through systemic mechanisms. Of note, all of N-glycosyltransferase genes that are expressed on RNA, protein, and glycan levels in both tissues are genetically regulated in only one of them (*B4GALT1, ST6GAL1, MGAT3*), or exhibit different glyQTLs in different tissues (*FUT8, FUT6*). Even thus, while a number of glycosyltransferases are expressed both in liver and lymphoid tissues, we provide evidence that the genetic variation regulating N-glycosylation in each of the two tissues is unique and does not overlap at the available resolution of the analysis. To the best of our knowledge, this is the first study to analyze and reveal the strong tissue-specificity of the genetic regulation of population variability of human protein N-glycosylation.

Further studies of the genetic regulation of N-glycosylation of individual proteins rather than bulk N-glycome will lead to the discovery of novel glyQTLs, which we cannot observe now due to a lack of power or noise in bulk N-glycome. Quantification of the N-glycome of purified proteins like IgG[40], TF[42], IgA[79,80], and apolipoprotein CIII[81], and other proteins, will be highly relevant to understanding the etiology of such diseases, as rheumatoid arthritis, hepatocellular carcinoma, IgA nephropathy, endocarditis[7]. Alternatively, the development and application of computational deconvolution approaches may be similar to those applied for bulk RNA-Seq[82].

In conclusion, our study offers insight into the genetic factors influencing blood plasma N-glycome, and, for the first time, establishes a genetic link between N-glycosylation, liver disease, and anti-inflammatory proteins. The identification of novel genes associated with metabolic and liver disease and N-glycosylation contributes to a deeper understanding of shared biological mechanisms and will facilitate future biomarker discovery and interpretation.

## Methods
We conducted a multicenter study using data from seven studies – TwinsUK ($N = 3918$), EPIC-Potsdam ($N = 2192$), PainOmics ($N = 1873$), SOCCS ($N = 1742$), SABRE ($N = 544$), QMDiab ($N = 325$), and CEDAR ($N = 170$) with a total sample size $N = 10,764$. Local research ethics committees approved all studies, and all participants gave written informed consent. The detailed description of the cohorts is shown in Supplementary Data 2 and Supplementary Notes.

### Glycome measurement and phenotype processing
Plasma N-glycome quantification of samples from all but SOCCS studies was performed at Genos Ltd using the protocol published previously[83]. Briefly, plasma N-glycans were enzymatically released from proteins by PNGase F, fluorescently labeled with 2-aminobenzamide and cleaned-up. Fluorescently labeled and purified N-glycans were separated by HILIC on a Waters BEH Glycan chromatography column. The fluorescence detector was set with excitation and emission wavelengths of 250 nm and 428 nm, respectively. Plasma N-glycome quantification for SOCCS samples was done at NIBRT by applying the same protocol with the only difference in the excitation wavelength (330 nm instead of 250 nm). Glycan peaks (GPs) – quantitative measurements of glycan levels – were defined by manual integration of intensity peaks in the chromatograms or were defined by automatic integration. The number of defined GPs varied among studies from 36 to 42, therefore, to conduct multi-center association analysis followed by meta-analysis, we harmonized the set of GPs by applying a recently published protocol[34] to a harmonized set of 36 GPs (Supplementary Data 17). To reduce experimental variation in glycan measurements, before genetic studies, raw glycan data were probabilistic median quotient normalized[84,85] and batch corrected centrally by the phenotype provider (Genos Ltd). More detailed information on glycan preprocessing can be found in the Supplementary Note. From the 36 directly measured glycan traits, 81 derived traits were calculated (Supplementary Data 1a). These derived traits average glycosylation features such as branching, galactosylation, and sialylation, etc. across different individual glycan structures and, consequently, they may be more closely related to individual enzymatic activity and underlying genetic polymorphism.

### Discovery and replication genetic association analysis
**Single trait association analysis.** Discovery genome-wide association studies were performed in (sub) cohorts of European descent: TwinsUK ($N = 2739$), EPIC-Potsdam ($N = 2192$), PainOmics ($N = 1873$), SOCCS (controls, $N = 459$) and SABRE ($N = 277$) with a combined sample size of 7540 (Supplementary Data 2b). Prior to GWAS, the total plasma N-glycome traits were adjusted for sex and age, and the residuals were quantile transformed to normal distribution. The genetic association analysis in each cohort was conducted using a similar protocol. We assumed an additive model of genetic effects. GWAS were based on the genotypes imputed from Haplotype Reference Consortium Results[86] or 1000 Genomes project[87]. Results of GWAS in each discovery cohort passed a strict quality control procedure followed by fixed-effects inverse-variance weighted meta-analysis. After quality control, 8.8 M SNPs were used for the downstream analysis.

To define genome-wide significant glyQTLs, we used a conventional genome-wide significance threshold, Bonferroni corrected by 28 independent glycan traits ($P \leq 1.79 \times 10^{-9}$) as suggested before[88]. We considered SNPs located in the same locus if they were located within 250 Kb from the leading SNP (the SNP with lowest $P$). Only the SNPs and the traits with the lowest $P$ are reported (leading SNP-trait pairs) in Table 1. The detailed procedure of locus definition is described in the Supplementary Note.

Replication GWAS were performed using (sub) cohorts of samples with European descent: SOCCS (colorectal cancer cases, $N = 1283$), TwinsUK ($N = 1179$), CEDAR ($N = 170$); South Asian descent: SABRE ($N = 267$) and Arabian, Indian, Filipino descent: QMDiab ($N = 325$) (Supplementary Data 2b). Results of GWAS in each discovery cohort passed a strict quality control procedure followed by fixed-effects inverse-variance weighted meta-analysis. The size of the replication sample ($N = 3224$, Supplementary Data 2b) was defined as to achieve 80% statistical power for a replication of the true association signal (Supplementary Note).

The genomic control inflation factor in the discovery GWAMA varied from 1.004 to 1.059. By contrast, an intercept of LD score regression[45] varied from 0.996 to 1.002 (Supplementary Data 5a), confirming minimal impact of genetic stratification on the GWAS

results. Hence, implementing Genomic Control correction in the analysis was unnecessary.

For replication of novel glyQTL, found at the discovery step, we used the leading SNP-trait pair that showed the most significant association. The replication threshold was set as $P < 0.05/28 = 0.00178$, where 28 is the number of replicated loci. Moreover, we checked whether the sign of estimated effect was concordant between discovery and replication studies.

**Identification of secondary associations in glyQTLs.** To identify secondary association signals at glyQTL in univariate analysis and capture the overall contribution to phenotypic variation, we performed conditional analysis using GCTA-COJO software, version 1.93.2beta[44]. This method uses summary-level statistics from a discovery meta-analysis and LD corrections between SNPs estimated from a reference sample for implementing a stepwise selection procedure including a series of conditional and joint regression analyses in which the SNP with the strongest association in the region is added to the regression model until no additional SNPs reach genome-wide significance. We used 1,429 unrelated individuals with European descent from SABRE cohort as reference samples for LD calculation. We used $P \leq 1.79 \times 10^{-9}$ as a genome-wide significance level and a default window setting to identify lead associations (Supplementary Data 4).

**Multi-trait association analysis.** To gain additional power of glyQTL detection, we performed a multivariate GWAS of total plasma N-glycome. It has been previously demonstrated that multivariate genetic association analysis of N-glycome, that is, a joint analysis of multiple N-glycome traits, has higher power for loci detection than a regression model under which glycome traits are analyzed independently of each other[28,41].

For discovery and replication analyses, we used discovery and replication GWAMA summary statistics, obtained in single-trait analysis. The discovery multivariate analysis was performed using the MANOVA-based method, adopted for analysis of a group of single-trait GWAS summary statistics (details are in Supplementary Note Discovery multivariate analysis)[28]. Discovery analysis was performed using the MultiABEL R package. Due to method requirements, we filtered out SNPs with sample size lower than 6,790 (which is 90% of 7540 samples). The statistical significance threshold for multivariate analysis was set at $P < 5 \times 10^{-8}/21$, where 21 is the number of multivariate traits described in Supplementary Data 1b and Supplementary Note. GlyQTLs with significant association were defined in the same way as for single-trait discovery.

For replication of multi-trait associated glyQTLs we used a complex four-step replication strategy as proposed by Ning et al. [29], which consists of the following steps: MANOVA, Phenotype Score, Pearson correlation method and Kendall correlation method. In the first step (MANOVA) we straightforwardly checked whether the locus is significantly associated with the multivariate trait in the replication cohort using the same test as in the discovery stage. The replication threshold was set as $P < \frac{0.05}{(7+16)} = 0.0021$, where 7 is the number of previously identified but not replicated loci and 16 is the number of novel loci. Then we checked whether the effect direction is consistent between the two cohorts, using the phenotype score approach[29]. Next, we evaluated the concordance of multivariate effect between two samples using Pearson and Kendall's correlation coefficients. We considered an association of the locus replicated if it had successfully passed MANOVA and phenotype score steps of replication. The multivariate effect of the locus replicated if it additionally had passed both Pearson's and Kendall's correlation steps of replication (Supplementary Note, Supplementary Data 3b, and Supplementary Fig. 3). Phenotype score-based replication was performed as in Shadrina et al. [41]. For each lead pair of SNP and trait group phenotype, we extracted coefficients of the linear combination of genotype onto multiple

atomic phenotypes, estimated for discovery cohort. We used them to construct the corresponding trait group phenotypes for further testing of an association between the lead SNPs and the derived linear combinations (see Supplementary Note for details). A locus was replicated if the association of the SNP with the constructed linear combination had the same direction of effect as in the discovery cohort and passed the threshold of $P < 0.0021$.

To evaluate the similarity between estimates of multivariate genetic effects from discovery and replication cohorts across multiple traits, we used an MC-based approach implemented in MultiABEL package (MV.cor.test() function)[28]. For both Pearson's and Kendall's correlation coefficients, we considered a multivariate effect for a specific SNP replicated if the 95% confidence intervals didn't include zero.

## SNP-based heritability and polygenic scores

SNP-based heritability was estimated using the LD Score regression software[45] embedded in the GWAS-MAP platform[89]. We used precomputed LD scores that were calculated from the European-ancestry samples in the 1000 Genomes Project. Only the 1,176,189 HapMap3 SNPs were included with a MAF > 0.05. For the purpose of heritability estimation and further post-GWAS analyses, we generated GWAMA summary statistics for the samples of European descent with N = 10,172. We used GWAMA summary statistics for the analysis in order to use the largest data set with homogeneous ancestry.

SBayesR method reweights the effect of each variant according to the marginal estimate of its effect size, the statistical strength of association, the degree of correlation between the variant and other variants nearby, and tuning parameters. This method requires a compatible LD matrix file computed using individual-level data from a reference population. For these analyses, we used publicly available shrunk sparse GCTB LD matrix including 1.1 million HapMap3 variants and computed from a random set of 50,000 individuals of European ancestry from the UKB data set[47,90]. SBayesR (gctb_2.03) was run for each chromosome separately, and with the default parameters except for the number of iterations (N = 5,000) and options for the stability of the algorithm (Supplementary Data 6). The prediction accuracy was defined as the proportion of the variance of a phenotype that is explained by PGS values (R2). To calculate PGS based on the PGS model, we used PLINK2 software[91], where PGS values were calculated as a weighted sum of allele counts. Out-of-sample prediction accuracy was evaluated using samples from the CEDAR cohort that were not used for discovery or replication.

## Prioritization of candidate genes in found loci

For the purpose of post-GWAS analyses, we generated GWAMA summary statistics for the samples of European descent ($N = 10,172$). GWAMA summary statistics passed the same QC procedure as discovery and replication GWAMA. We applied an ensemble of methods to prioritize plausible candidate genes in the loci with found and replicated glyQTL (32 in univariate and 8 in multivariate analysis). We applied eight approaches to prioritize the most likely effector genes: (1) prioritization of the nearest gene; (2) prioritization of genes with known role in biosynthesis of N-glycans; (3) genes of congenital disorders of glycosylation; (4) genes with direct experimental support for regulation of protein N-glycosylation; (5) prioritization of genes containing variants in strong LD ($R^2 \geq 0.8$) with the lead variant, which are protein truncating variants (annotated by Variant Effect Predictor, VEP[92]) or predicted to be damaging by FATHMM XF[93], FATHMM InDel[94]; (6) prioritization of genes whose eQTL and/or (7) pQTL are colocalized with glyQTL; (8) prioritization of genes based on the gene set and tissue/cell type enrichment, calculated by Data-driven Expression Prioritized Integration for Complex Traits (DEPICT) framework[31]. We prioritized the most likely 'causal gene' for each association using a consensus-based approach, selecting the gene with

the highest, unweighted sum of evidence across all eight predictors. In the case of equality of the scores for two genes, we prioritized both genes, but we selected the random one to represent the locus in the tables/figures to avoid cluttering.

**Functional annotation of genetic variants.** We inferred the possible molecular consequences of genetic variants in glyQTLs. We focused on variants in LD with lead (for univariate and multivariate signals) and sentinel variants (for univariate signals) picked by COJO. We created a "long list" of putative causal variants using PLINK version 1.9 (--show-tags option), applied to whole genome re-sequenced data for 503 European ancestry individuals (1000 Genomes phase 3 version 5 data). The size of the window to find the LD in both cases was equal to 500 kb. The default value of $R^2 > 0.8$ was taken as a threshold to include SNPs into the credible sets. Ensembl Variant Effect Predictor (VEP) (Supplementary Data 7d) and by FATHMM-XF (Supplementary Data 7b), FATHMM-INDEL (Supplementary Data 7c) to reveal pathogenic point mutations.

**Genes of N-glycan biosynthesis and Congenital Disorders of Glycosylation.** We searched for the genes encoding glycosyltransferases – enzymes, with a known role in N-glycan biosynthesis[95], located in the ±250 Kb-vicinity of the lead SNPs in glyQTLs. Additionally, we prioritized genes with known mutations, that cause Congenital Disorder of Glycosylation according to MedGen database (https://www.ncbi.nlm.nih.gov/medgen/76469) that are located in the vicinity of ±250 kb from the lead SNPs.

**Colocalization with eQTL and pQTL.** To find potential pleiotropic effects of glyQTL on gene expression levels in relevant tissues, we applied Summary data-based Mendelian Randomization (SMR) analysis followed by the Heterogeneity in Dependent Instruments (HEIDI)[32] on expression of quantitative trait loci (eQTLs) obtained from Westra Blood eQTL collection[96] (peripheral blood), GTEx (version 7) eQTL collection[97] (liver, whole blood), CEDAR eQTL collection[49] (CD19 + B lymphocytes, CD8 + T lymphocytes, CD4 + T lymphocytes, CD14+ monocytes, CD15+ granulocytes) and on protein quantitative trait loci (pQTLs) using SomaLogic datasets[98,99]. As outcome variable we used univariate association results for the N-glycome trait with the most significant association; in the case of glyQTLs replicated only in multivariate analysis, we used summary statistics for the most associated univariate trait as the primary trait in the analysis. For gene prioritization, we focused only on cis-eQTL and cis-pQTL, defining them as associations located within 1 Mbp of the nearest gene boundary.

The results of the SMR test were considered statistically significant if $P_{adj} < 0.05$ (Benjamini-Hochberg adjusted $P$). The significance threshold for HEIDI tests was set at $P = 0.05$ ($P < 0.05$ corresponds to the rejection of the pleiotropy hypothesis) (Supplementary Data 7e, 7f).

**DEPICT.** Gene prioritization and gene set and tissue/cell type enrichment analyses were performed using the Data-driven Expression Prioritized Integration for Complex Traits framework (DEPICT)[31]. DEPICT analysis was conducted for SNPs associated with any N-glycosylation trait at $P < 5 \times 10^{-8}/28$ in univariate analysis and with any N-glycosylation trait group at $P < 5 \times 10^{-8}/21$ in multivariate analysis. The significance threshold for DEPICT analysis was set at False Discovery Rate $FDR < 0.20$ (Supplementary Data 7g, 7h, 7i).

**Colocalization with TF and IgG**
In this study, colocalization analysis (SMR-θ)[49] was conducted for total plasma, IgG and TF glyQTLs. The analysis was restricted to loci that were a) previously implicated in TF GWAS[42] (4 loci); IgG GWAS[40] (15 loci); both (2 loci) (Supplementary Data 8b), b) reached genome-wide significance in the GWAMA of European descent ($N = 10,172$), c)

replicated in this study. Statistic θ is a weighted correlation, whose computation requires information on p-values and effect direction. The high absolute value (e.g. $|\theta| > 0.7$) means the locus likely has a pleiotropic effect on investigated traits.

**Pleiotropy with disease**
To study potential pleiotropic effects on a range of traits associated with various medical conditions SMR/HEIDI analysis was carried out similarly to that for colocalization with eQTL and pQTL.

Summary statistics for complex and medical conditions-related traits were obtained from the UK Biobank[100], the CARDIoGRAM Consortium (http://www.cardiogramplusc4d.org/), the Psychiatric Genomics consortium (https://pgc.unc.edu/) and other trait collections from other studies (see Supplementary Data 10 for the full list of the traits analyzed). We conducted analysis separately for the disease-related traits and other complex traits.

**Associations between PGS for plasma N-glycosylation traits and disease phenotypes**
To test the associations between the 117 human plasma N-glycosylation traits and ICD-10 disease phenotypes, we used logistic regression considering PGS for each glycan trait as a predictor for each disease phenotype in turn.

The list of diseases was taken from medical histories and questionnaires obtained from non-related UK Biobank participants of European descent for which we had PGS for N-glycosylation traits calculated ($N = 374,303$). All medical codes were preliminary filtered by prevalence (> 0.5% and < 99.5%). For this analysis, we used 167 groups of codes that fall into Chapters I-XV of the UK Biobank classification of phenotypes. These codes describe a wide range of phenotypes, including infectious diseases, endocrine, nutritional, and metabolic diseases, diseases of the nervous system, diseases of the circulatory, respiratory, digestive, and other systems, etc.

To perform logistic regression analyses we used the standard glm() function in R v.4.2.2. programming language. We included sex, age, batch number, and the first ten principal components of the kinship matrix (PC 1-10) as covariates in addition to the PGS predictor. Finally, we filtered out the results not passing the significance threshold for the association of $P < 0.05/(28 \times 167) = 1.07 \times 10^{-5}$, where 28 is the number of plasma N-glycome principal components explaining over 99% of the 117 N-glycosylation traits variation, and 167 is the numbers of ICD-10 codes.

**Mendelian randomization and sensitivity analyses**
In the previous step, we identified 64 pairs of associated disease phenotypes and plasma N-glycosylation traits. To investigate the causal relationships between these traits we performed a bidirectional two-sample MR analysis[33]: for each pair, we performed two MR analyses using the glycosylation trait as exposure and the disease as outcome and vice versa.

As the sources of the summary statistics for MR analyses, we used the largest available GWAMA for plasma N-glycosylation traits in a cohort of European descent described previously in the current study ($N = 10,172$) and GWAS available from the UK Biobank database (for more details about these cohorts see Supplementary Data 15).

The framework of the two-sample MR was specified before the analysis. Genetic IVs for the two-sample MR were identified as follows. First, the set of SNPs present in both the GWAS for the exposure and outcome traits was selected. Then, for this overlapping set of SNPs in the GWAS for the exposure trait, we performed clumping for independence using PLINK2[91] within a 10,000 kb window. Additional parameters for clumping included an $r^2 > 0.001$ threshold for correlation, IVs with minor allele frequency $MAF < 0.05$ were excluded. When plasma N-glycosylation traits were considered as exposures, the $P$ threshold for clumping was defined as $5 \times 10^{-8}/28 = 1.79 \times 10^{-9}$ (28 -

number of plasma N-glycome principal components explaining over 99% of the 117 N-glycosylation traits variation). When the disease phenotypes were considered as exposure, this threshold was set at $5 \times 10^{-8}/14 = 3.57 \times 10^{-9}$ (14 - number of disease phenotypes significantly associated with at least one plasma N-glycosylation trait in the logistic regression analysis).

Summary statistics for IVs in the exposure and outcome GWAS data were processed using the TwoSampleMR R package[33]: the data were harmonized, excluding ambiguous/triallelic SNPs. Only the pairs where at least 2 IVs were available for the exposure trait were considered for further analysis. MR analysis was performed using mr_report() function from the TwoSampleMR R package. Significance thresholds for the MR results were set as $\frac{0.05}{49} = 0.001$ for the analysis of 49 traits pairs where glycans were considered as exposures, and as $\frac{0.05}{64} = 0.00078$ for the analysis of the 64 pairs where diseases were considered as exposures. If at least one of the MR methods used (Inverse variance weighted, MR Egger, Weighted median, Weighted mode, or Simple mode) produced a statistically significant causal estimate, that pair of traits was selected for the follow-up sensitivity analyses.

Follow-up sensitivity analyses included those automatically implemented in the mr_report() function, such as heterogeneity tests, test for directional horizontal pleiotropy, leave-one-out analysis, forest plot, and funnel plot.

In addition to the sensitivity analyses described above, for the pairs where plasma N-glycosylation traits were used as exposures, since the number of IVs was very low (2-3 SNPs), we performed colocalization analysis (SMR-HEIDI) for each of the IVs. The results of the SMR test were considered statistically significant if Benjamini-Hochberg adjusted $P < 0.05$. The significance threshold for HEIDI tests was set at $P = 0.05$ which corresponds to the rejection of the pleiotropy hypothesis.

For the pairs where the disease was the upstream exposure trait (disorders of lipoprotein metabolism and other lipidaemias, E78) and 27 IVs were available, we identified pleiotropic IVs using MR-PRESSO R package, detecting the IVs for which $P$ of the test for outliers in MR-PRESSO were <1. Then we repeated the MR analysis as described above excluding these SNPs.

To sum it up, for an effect of exposure on an outcome to be considered consistent with a causal relationship, 2 criteria must be met. Firstly, at least one of the MR methods used in the primary analysis must produce an estimate of causal association that exceeds the Bonferroni-corrected significance thresholds. The second criterion for the effects of diseases on glycan traits was that the direction and magnitude of the effects estimated after the removal of pleiotropic IVs were consistent with the primary analyses. In the case of the effects of glycan traits on diseases, where only 2-3 IVs were available, we designated the second criterion as significant association in the SMR test together with a non-significant HEIDI test for pleiotropy for at least one of the available IVs.

**Reporting summary**

Further information on research design is available in the Nature Portfolio Reporting Summary linked to this article.

## Data availability

The full genome-wide summary association statistics for 117 glycome traits from the European GWAMA (Genome-Wide Association Meta-Analysis) conducted on participants of European ancestry, totaling 10,172 individuals, have been deposited in the Zenodo database under accession code 15057709 (CC BY 4.0). (https://doi.org/10.5281/zenodo.15057709). The full genome-wide summary association statistics for 117 glycome traits from the discovery GWAMA (Genome-Wide Association Meta-Analysis) conducted on participants, totaling 7,540 individuals, have been deposited in the Zenodo database under accession code 15150419 (CC BY 4.0) (https://doi.org/10.5281/zenodo.15150419). The first part of the full genome-wide summary association statistics for 117 glycome traits from the replication GWAMA (Genome-Wide Association Meta-Analysis) conducted on participants, totaling 3,224 individuals, have been deposited in the Zenodo database under accession code 15161945 (CC BY 4.0) (https://doi.org/10.5281/zenodo.15161945). The second part of the full genome-wide summary association statistics for 117 glycome traits from the replication GWAMA (Genome-Wide Association Meta-Analysis) conducted on participants, totaling 3,224 individuals, have been deposited in the Zenodo database under accession code 15166735 (CC BY 4.0) (https://doi.org/10.5281/zenodo.15166735). The data generated in the secondary analyses of this study are included with this article in the Supplementary Data. Source data are provided with this paper.

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

## Acknowledgements

The work of S.Sh., A.T., D.M., A.S., and Y.S.A. was supported by the Research Program at the Moscow State University (MSU) Institute for Artificial Intelligence. The study was conducted using the UK Biobank resource under application #59345. The work of E.E., Y.A.T. was supported by the budget project of the Institute of Cytology and Genetics FWNR-2022-0020. European Community's Seventh Framework Programme funded project PainOmics (602736). TwinsUK is funded by the Wellcome Trust, Medical Research Council, Versus Arthritis, European Union Horizon 2020, Chronic Disease Research Foundation (CDRF), Zoe Ltd and the National Institute for Health Research (NIHR) Clinical

Research Network (CRN) and Biomedical Research Centre based at Guy's and St Thomas' NHS Foundation Trust in partnership with King's College London. The TwinsUK Study was approved by London-Westminster Research Ethics Committee (REC reference EC04/015), and Guy's and St Thomas' NHS Foundation Trust Research and Development (R&D). The TwinsUK BioBank was approved by the HRA - Liverpool East Research Ethics Committee (REC reference 19/NW/0187), IRAS ID 258513. Glycan analysis performed in Genos was supported by Horizon Europe grants GlycanSwitch (ERC Synergy grant # 101071386), INITIALIZE (grant # 101094099) and SynHealth (grant #101159018). All participants provide written, informed consent. We thank Toma Keser, Mirna Šimurina, Marija Vilaj, Jerko Štambuk, Ivan Gudelj, Thomas S. Klarić, Jasminka Krištić, Jelena Šimunović, Julija Jurić, Ana Momčilović, Najda Rudman, and Maja Hanić for their assistance with glycan analysis. We thank Dmitry Shtokalo for valuable discussion and the Federal Research Center for Information and Computational Technologies SB RAS (FRC ICT SB RAS) for assistance with computational resources.

## Author contributions

S.Sh. coordinated this study; S.Sh., A.T., O.Z., D.M., A.S., E.E., E.T., A.N., S.F., N.A.P., Y.A.T. contributed to the design of the study, carried out statistical analysis; A.T., D.M., A.S., E.T., O.Z., S.Sh. produced the figures; S.Sh., A.T., O.Z., D.M., A.S., Y.A.T., contributed to interpretation of the results; S.Sh., A.T., O.Z., D.M., A.S., A.N. and Y.S.A. wrote the first version of the manuscript; S.Sh., A.T., O.Z., D.M., A.S., and Y.S.A. wrote the revised second version of the manuscript; S.Sh., E.T., E.E., S.F., V.V. and Y.A.T. contributed to data harmonization and quality control; F.V., I.T.-A., T.Š. contributed to plasma N-glycome measurements and quality contol; M.M., T.S. analyzed TwinsUK dataset and contributed to interpretation of the results; M.T., M.D. analyzed SOCCS dataset and contributed to interpretation of the results; L.K., F.W., D.P., J. Van Z., M.A. designed PainOmics study and contributed to interpretation of the results; K.S. analyzed QMDiab dataset and contributed to interpretation of the results; M.G. designed CEDAR study and contributed to interpretation of the results; N.Ch. designed SABRE study and contributed to interpretation of the results; C.W. and M.B.S. contributed to the data collection and analyses of EPIC-Potsdam and to the interpretation of results; Y.S.A. and G.L. conceived and oversaw the study, contributed to the design and interpretation of the results; all co-authors contributed to the final manuscript revision.

## Competing interests

Y.S.A. is a full-time employee of GSK PLC and receives a salary and stock options as compensation. G.L. is a founder and owner of Genos Ltd, a biotech company that specializes in glycan analysis and has several patents in the field. O.Z., T.Š., F.V., and I.T.-A are employees of Genos Ltd. All other authors declare no competing interests.

## Additional information

[1]MSU Institute for Artificial Intelligence, Lomonosov Moscow State University, Moscow, Russia. [2]Genos Glycoscience Research Laboratory, Borongajska cesta 83H, Zagreb, Croatia. [3]Institute of Cytology and Genetics, Novosibirsk, Russia. [4]Novosibirsk State University, Novosibirsk, Russia. [5]Department of Twin Research and Genetic Epidemiology, School of Life Course Sciences, King's College London, St Thomas' Campus, Lambeth Palace Road, London, United Kingdom. [6]NIHR Biomedical Research Centre at Guy's and St Thomas' Foundation Trust, London, United Kingdom. [7]Department of Molecular Epidemiology, German Institute of Human Nutrition Potsdam-Rehbruecke, Nuthetal, Germany. [8]PolyOmica, 's-Hertogenbosch, Hertogenbosch, PA, The Netherlands. [9]Colon Cancer Genetics Group, Cancer Research UK Scotland Centre, Institute of Genetics & Cancer, Western General Hospital, The University of Edinburgh, Edinburgh, United Kingdom. [10]D-IAS, Danish Institute for Advanced Study, Department of Public Health, University of Southern Denmark, J.B. Winsløws Vej 9, Odense C, Denmark. [11]Engelhardt Institute of Molecular Biology, Russian Academy of Sciences, Moscow, Russia. [12]Vavilov Institute of General Genetics Russian Academy of Sciences, Moscow, Russia. [13]St. Catherine Specialty Hospital, Ulica Kneza Branimira, 71E Zagreb, Croatia. [14]University of Split School of Medicine, Split, Croatia. [15]University of Osijek School of Medicine, Osijek, Croatia. [16]Department of Anesthesiology and Multidisciplinary Paincentre, ZOL, Genk, Lanaken, Belgium. [17]Department of Anesthesiology and Pain Medicine, Maastricht University Medical Centre, P. Debyelaan 25, Maastricht, The Netherlands. [18]Unit of Animal Genomics, WELBIO, GIGA-R and Faculty of Veterinary Medicine, University of Liège, (B34) 1 Avenue de l'Hôpital, Liège, Belgium. [19]Department of Physiology and Biophysics, Weill Cornell Medicine-Qatar, Education City, Doha, Qatar. [20]Centre Lemanique d'antalgie et neuromodulation – EHC, Morges, Switzerland. [21]MRC Unit for Lifelong Health & Ageing University College London, London, United Kingdom. [22]Department of Molecular Epidemiology, German Institute of Human Nutrition Potsdam- Rehbruecke, 14558 Nuthetal, Germany. [23]German Center for Diabetes Research (DZD), Neuherberg, Germany. [24]Institute of Nutrition Science, University of Potsdam, Potsdam, Germany. [25]Wellcome Sanger Institute, Cambridge, United Kingdom. [26]University of Zagreb Faculty of Pharmacy and Biochemistry, Zagreb, Croatia. ✉e-mail: yurii@bionet.nsc.ru

