## [Transparent Peer Review file · Nature Communications]

A genome-wide association study in 10,000 individuals links plasma N-glycome to liver disease and anti-inflammatory proteins

Corresponding Author: Dr Sodbo Sharapov

Version 0:

Reviewer comments:

Reviewer #1

(Remarks to the Author)

In this study, the authors investigated the heritability and genetic architecture of N-glycosylation traits collected from the blood proteome. To achieve this, meta-analyses of previously published GWAS were performed to achieve a total sample size of 10,000 samples. No additional N-glycosylation data collection was performed. The analyses in this manuscript included estimates of heritability, GWAS, polygenic score analyses, colocalization to identify tissue-specific consequences of glyQTLs, and mendelian randomization. The key findings include novel genes linked to N-glycosylation and tissue-specific information about variants linked to N-glycosylation. The breadth of data described in the manuscript underscores the amount of effort that went into the current draft. However, the potential link to human health related to these findings is not strongly supported by polygenic score analysis or MR. The small size of the study population and the lack of a compelling link to human health diminish its potential impact. The stated conclusion is “establishes a genetic link between N-glycosylation, liver disease, and anti-inflammatory proteins,” but this point was not fully articulated in the manuscript. The conclusions may be overstated and the manuscript would benefit from the conclusion being toned down to more accurately reflect the results.

High level criticism:

Results are overwrought with repetition and technical details. It is a difficult read and hard to identify key points. The manuscript could benefit from moving some analyses to the supplement.

There are a lot of grammatical errors throughout that sometimes change the intended meaning of statements.

Specific points:

Add to the introduction: background about whether it's difficult or expensive to get glycan levels. Is it lack of interest, knowledge, or cost that is causing such a low sample size across the literature. Or are glycans only recently being recognized as important?

Line 78: I think “biomarkers” needs a clarifying statement: “AN essential part of therapeutics, as well as biomarkers for X, Y, and Z”

Line 140: “genetically predicted glycan levels”. These are glycan polygenic scores, not genetically-predicted glycan levels. Many figures (Fig 1b, Fig 2a) and tables are out of order. For example, Supplemental Table 3 is the first table mentioned in the text. I also don't usually see panels (e.g. A,B,C) for Tables. I would think they should either be separate tables or combined into one.

Line 157: It looks like the $N = 7,540$ number is the number of total participants in the training set as opposed to the European subset as stated, because $7540 + 3224$ (replication set) = your total participant count (10764).

Line 239: I don't know what “for the latter” is referring to, and “previous works and reviews” requires more than one reference. The figures overall are well put together. For Figure 3, I would recommend some visual indication that 3b is only the diseases from 3a, such as a bracket. Why do the genes in 3b not follow the same clustering order as 3a?

The PGS results are never put into the greater context. Its concordance and discordance with other results (e.g. MR) should be explained and interpreted in the Discussion. Alternatively, if the PGS results are not important enough to mention in the Discussion, the manuscript may benefit from removing this section entirely, or moving to the Supplement.

MR analysis: There is a lot of pleiotropy here, and horizontal pleiotropy can strongly influence MR outcomes. Should the two-sample MR after removal of pleiotropic IVs be the primary analysis rather than a sensitivity analysis? There is very little left after removal of horizontal pleiotropic IVs.

Line 432: "we demonstrate, for the first time, that specific genes are associated with both N-glycosylation and liver disease, offering a starting point for genetically-guided investigation of the functional mechanisms of this phenotypic association." I thought that this was one of the strongest statements in the manuscript and should be elaborated on and more clearly articulated throughout.

Line 451: Can you comment on whether or how the horizontal pleiotropy test can help to distinguish between these two hypotheses?

Reviewer #2

(Remarks to the Author)

Sharapov et al. present the largest genome-wide association study to date of plasma protein N-glycosylation, including 117 univariate and 21 multivariate glycosylation traits and identifying 16 novel loci compared to previous studies. The authors prioritise the most likely effector genes in each loci and describe them with regard to their glycosylation patterns. They show that glyQTLs colocalize with GWAS signals for various diseases and traits, and that genetically predicted glycosylation is associated with disease outcomes in UKBB. Finally, they use bi-directional Mendelian Randomization to highlight some potentially causal associations between glycosylation traits and diseases. The work is extensive and the authors have done good job of highlighting the novelty of their results, providing new insights into the relationships between candidate genes, N-glycosylation and disease. The methods are appropriate, descriptions are detailed and a lot of additional information is provided in supplementary data.

My specific comments are as follows:

1. For the gene prioritization - how was the primary gene selected if multiple genes had the same score?
2. For the colocalization with eQTLs and pQTLs, please clarify how a gene was prioritized if the glyQTL colocalized with multiple genes and proteins? For example in ST-6b, the SNP rs1260326 is annotated with eQTL NRBP1 while in ST-6f it seems to colocalize with 7 different genes. Similarly, the SNP rs8283 annotated to pQTL for C4 in ST-6b colocalizes with 6 proteins in table ST-6g. It would also be helpful to add cis/trans information to the pQTL table (ST-6g).
3. The expression of candidate genes is shown for hepatocytes and plasma cells (Fig 2a), which is valid given established knowledge of glycosylation of plasma proteins, but did the authors perform any formal tissue specificity enrichment analysis of the GWAS results? The authors state that the results indicate strong tissue-specificity of N-glycosylation in liver and lymphoid tissue but it seems that these tissues were preselected.
4. A big part of the results describes colocalization of glyQTLs with disease loci, PGS associations with diseases and MR analysis but these are not really referred to in the discussion. What could be the mechanisms at these loci? Is it likely that the disease risk is mediated through glycosylation or the other way around? The MR results indicate some bi-directional relationships that could be discussed.
5. In the discussion, the authors suggest that colocalization between glyQTLs and cis-pQTLs could indicate either changes in protein levels or changes in SomaScan binding affinity due to glycosylation changes. Could the latter be even more important mechanisms when considering trans-pQTLs? Many of the colocalized loci in ST-6g are established trans pQTL hotspots observed in proteomic GWAS studies, including ABO, CHF and SERPINA1.
6. It would be helpful to highlight parts of the supplementary tables to link better with the findings in the main text
 - a. The 40 novel loci could be better indicated in supplementary tables 5a-b
 - b. Lines 174-184: the text describes nine loci with secondary signals, could these be highlighted in suppl table 6a?
 - c. Lines 225-228: it is difficult to see how the 2 colocalized proteins and 10 genes mentioned here are selected from ST 6g-f. Maybe these could be highlighted and/or the reader here also referred to ST-6b. It would also be helpful for the pQTL table if cis and trans pQTLs were indicated.

Minor comments:

- Fig1a: The scale of the Y axis is unclear. Are these $-\log_{10}(P\text{-values})$ in the outer circles? number of associations in the middle? Please clarify.
- P.5, line 160 – Wrong table reference, should this be Supplementary Table 3b?
- P7, line 244: HRP should be HPR
- Can you please specify what the columns and heatmap values in Supplementary Figure 3 reflect? (variants and $|\theta|$ values?)
- I think not all Supplementary Tables are referred to nor in the correct order

Reviewer #3

(Remarks to the Author)

Version 1:

Reviewer comments:

Reviewer #1

(Remarks to the Author)

All of our concerns have been addressed in these revisions.

Reviewer #2

(Remarks to the Author)

Overall the authors have addressed my comments.

However, I have a couple of additional comments regarding the newly added bi-directional MR analyses in Supplementary Tables 16a-b. First, these should be referred to in the Results and not only in the Discussion. Second, it is difficult to link the text in Discussion to the results in ST16a without the exposure number (PGPxx) being indicated in the text or the description (A3G3S3) in the table. The authors state that they do not find causal associations for ALP or GGT – yet such associations are shown in ST16a? Do they mean for specific exposures? This is unclear.

Please check phrasing of results in the MR/PGS discussion – especially this sentence „MR indicated positive causal links between disorders of lipoprotein metabolism and PGS for M9 and Mtotal glycans, and between PGS for M6 glycan and asthma.“

In my opinion, it would be more appropriate to say that the MR analysis indicates a causal effect of x (M9 and Mtotal glycans) on y (lipoprotein metabolism). This should also be considered in the Results.

Also, please check the spelling overall [for example “abundance”, “instance”, “links relationship”, “and. negative”].

Reviewer #3

(Remarks to the Author)

Responses to the reviewers

Reviewer: 1

Remarks to the Author

1. *In this study, the authors investigated the heritability and genetic architecture of N-glycosylation traits collected from the blood proteome. To achieve this, meta-analyses of previously published GWAS were performed to achieve a total sample size of 10,000 samples. No additional N-glycosylation data collection was performed. The analyses in this manuscript included estimates of heritability, GWAS, polygenic score analyses, colocalization to identify tissue-specific consequences of glyQTLs, and mendelian randomization. The key findings include novel genes linked to N-glycosylation and tissue-specific information about variants linked to N-glycosylation. The breadth of data described in the manuscript underscores the amount of effort that went into the current draft.*

Response: Thank you very much for your comments and evaluation of our submission.

2. *However, the potential link to human health related to these findings is not strongly supported by polygenic score analysis or MR. The small size of the study population and the lack of a compelling link to human health diminish its potential impact. The stated conclusion is “establishes a genetic link between N-glycosylation, liver disease, and anti-inflammatory proteins,” but this point was not fully articulated in the manuscript. The conclusions may be overstated and the manuscript would benefit from the conclusion being toned down to more accurately reflect the results.*

Response: Thank you for your comments. In the revised manuscript (see Lines 472-475), we have strengthened our claims by performing bidirectional Mendelian randomization (MR) analyses to further explore the genetic link between blood plasma N-glycosylation and markers of liver physiology (see Supplementary Table 16a) and extended the paragraph in Discussion (see Lines 460-475). Specifically, we investigated causal relationships between all measured N-glycan levels and non-alcoholic fatty liver disease (NAFLD) —a common chronic liver disease—as well as four clinically relevant liver biomarkers: blood levels of alanine aminotransferase (ALT), alkaline phosphatase (ALP), gamma-glutamyl transferase (GGT), and aspartate aminotransferase (AST), which reflect distinct aspects of liver physiology and health.

Our MR results demonstrate that increased levels of glycans PGP25 and PGP27 (both possessing the tri-sialylated, tri-antennary A3G3S3 structure), predominantly present on alpha-1-acid glycoprotein, alpha-1-antitrypsin, and beta-2-glycoprotein 1 (primarily hepatocyte-derived proteins) [<https://doi.org/10.1016/j.eng.2023.03.013>], are significantly associated with lower AST levels. These glycoproteins play critical roles in modulating systemic inflammation, oxidative stress, and immune responses, processes directly relevant to liver injury and repair [<https://doi.org/10.4254/wjh.v13.i11.1688>]. Therefore, the association between increased A3G3S3 glycan levels and lower AST might reflect a protective effect against hepatocellular damage or stress.

However, we did not observe significant causal associations for ALT, ALP, GGT, or NAFLD. This lack of significant findings may result partly from limited

statistical power due to the small number of genetic instrumental variables available for glycan traits (3–8 instruments per trait). Alternatively, there might be a specific biological effect linking these glycans to AST levels, but not necessarily to other liver enzymes or NAFLD. Consequently, the absence of significant findings for these other liver phenotypes should be interpreted with caution.

We have adjusted our conclusions (Lines 472-475) to reflect that while our data robustly supports a genetic association between the A3G3S3 glycan structure and AST, further studies with enhanced statistical power are necessary to clarify potential broader implications for liver health.

High level criticism:

1. *Results are overwrought with repetition and technical details. It is a difficult read and hard to identify key points. The manuscript could benefit from moving some analyses to the supplement. There are a lot of grammatical errors throughout that sometimes change the intended meaning of statements.*

Response: Thank you for your suggestion. We have revised the Results section, removed repetitions (Lines 182-184, 217-220), and moved the technical details (Lines 166-173, 244-247) to the Methods section. We believe that these changes increased the readability and understanding of the manuscript.

Specific points:

2. *Add to the introduction: background about whether it's difficult or expensive to get glycan levels. Is it lack of interest, knowledge, or cost that is causing such a low sample size across the literature. Or are glycans only recently being recognized as important?*

Response: Thank you for this suggestion. In response, we have revised the sentence “Abundance of total plasma N-glycans can be quantified through various analytical methods” to “By the beginning of the 2010s, advancements in high-throughput methods for N-glycome profiling facilitated research into the genetic control of N-glycosylation” (see Lines 113-114). This change aims to clarify the evolution of methodologies in glycan analysis and explain the low sample size across the literature.

3. *Line 78: I think “biomarkers” needs a clarifying statement: “AN essential part of therapeutics, as well as biomarkers for X, Y, and Z”*

Response: Thank you for your comment regarding the clarification of the term "biomarkers" in Lines 82-83. We revised our statement and added supporting references. The updated sentence now reads:

"Glycans are considered to be potential therapeutic targets¹⁰⁻¹², essential part of therapeutics¹³⁻¹⁵, as well as biomarkers for cancer, diabetes, inflammatory and autoimmune disease¹⁶⁻¹⁸, which makes glycobiology a promising field for future clinical applications."

4. *Line 140: “genetically predicted glycan levels”. These are glycan polygenic scores, not genetically-predicted glycan levels.*

Response: Thank you for pointing this out. We have revised sentences in Lines 145 and 201-202 by substituting “genetically predicted glycan levels” with “glycan polygenic scores” for improved clarity and accuracy.

5. *Many figures (Fig 1b, Fig 2a) and tables are out of order. For example, Supplemental Table 3 is the first table mentioned in the text. I also don't usually see panels (e.g. A,B,C) for Tables. I would think they should either be separate tables or combined into one.*

Response: Thank you very much for pointing this. We have corrected the order of the Supplementary Tables and Figures. Additionally, we combined some Supplementary Tables for convenience.

6. *Line 157: It looks like the $N = 7,540$ number is the number of total participants in the training set as opposed to the European subset as stated, because $7540+3224$ (replication set) = your total participant count (10764).*

Response: Thank you for the comment. We revised the text in line 161-162 to accurately reflect that the $N = 7,540$ encompasses the total number of participants in the training (GWAS discovery) set. The updated sentence now reads: "We then performed a fixed-effect discovery meta-analysis of the subcohorts, which comprised a total of 7,540 participants."

7. *Line 239: I don't know what “for the latter” is referring to, and “previous works and reviews” requires more than one reference.*

Response: Thank you for the comment. We have revised the sentence to avoid uncertainty, and now it reads (line 251) “In the following discussion, we focus on thirteen novel candidate genes that were not yet confidently identified in previously published GWASs of human protein N-glycosylation; for information on those earlier studies, we refer the reader to prior works and published reviews.” We also added relevant references number ^{25,26,36,37,40,41}.

8. *The figures overall are well put together. For Figure 3, I would recommend some visual indication that 3b is only the diseases from 3a, such as a bracket. Why do the genes in 3b not follow the same clustering order as 3a?*

Response: Thank you for your feedback. We have added a bracket into a figure as a visual indication that Fig. 3b shows diseases present in the “Diseases” subsection of the Fig. 3a and ensured that the clustering remains consistent.

9. *The PGS results are never put into the greater context. Its concordance and discordance with other results (e.g. MR) should be explained and interpreted in the Discussion. Alternatively, if the PGS results are not important enough to mention in the Discussion, the manuscript may benefit from removing this section entirely, or moving to the Supplement.*

Response: Thank you for your feedback. We have revised the discussion to include the PGS and MR results (Lines 444-465)

“We found positive associations between cardiovascular disease and PGS for high-mannose glycans, previously suggested as biomarkers for cardiovascular events in patients with diabetes [57], and negative associations between metabolic disorders and PGS for galactosylated glycans. MR indicated positive causal links relationship between disorders of lipoprotein metabolism and PGS for M9 and Mtotal glycans, and between PGS for M6 glycan and asthma. Elevated high mannose glycans (e.g. M9, M6) may reflect disrupted glycan processing in the Golgi apparatus [58,59], a common feature in various pathological conditions including cancer [60]. Alternatively, their increase plasma abundance could result from elevated levels of specific carrier proteins. For instance, apolipoprotein B-100 [2], abundant in high-mannose structures (M9) [2], is closely linked to lipid disorders and cardiovascular disease [61–66], whereas elevated IgE antibodies, which prominently carry M6 glycans [2], play a role in allergic inflammation and asthma [67]”.

Additionally, we discussed the results of colocalization analysis between glyQTLs and pQTLs and their possible mechanism in the Lines (500-510); and the results of MR between N-glycans and blood levels of liver enzymes - alanine aminotransferase (ALT), alkaline phosphatase (ALP), gamma-glutamyl transferase (GGT), and aspartate aminotransferase (AST) – in the Lines 460-475.

10. MR analysis: There is a lot of pleiotropy here, and horizontal pleiotropy can strongly influence MR outcomes. Should the two-sample MR after removal of pleiotropic IVs be the primary analysis rather than a sensitivity analysis? There is very little left after removal of horizontal pleiotropic IVs.

Response: Thank you for the comment. To make the criteria for assessment of the evidence for causal relationships in our data clearer, we have now added a paragraph in Lines 846-854 explicitly describing them:

“To sum it up, we asked the effect of an exposure on an outcome to meet two criteria to be considered consistent with a causal relationship. Firstly, at least one of the MR methods used in the primary analysis must produce an estimate of causal association that exceeds the Bonferroni-corrected significance thresholds. Secondly, the direction and magnitude of the effects of diseases on glycan traits estimated after the removal of pleiotropic IVs were asked to be consistent with the primary analyses. In the case of the effects of glycan traits on diseases, where only 2-3 IVs were available, we designated the second criterion as a significant association in the SMR test together with a non-significant HEIDI test for pleiotropy for at least one of the available IVs.”

As you can see, the results for MR analysis with diseases as outcome and glycan traits as exposures, are considered to be consistent with causality only if the effects hold after we remove the pleiotropic SNPs using MR-PRESSO. The disappearance of pleiotropy is additionally controlled by the tests for heterogeneity and for the significance of the MrEgger intercept. Both tests are included in the standard MR protocol implemented in the TwoSampleMR R package. The results of these analyses can be found in the Supplementary Table ST-11b and ST-11c, respectively.

We do not use the two-sample MR after removal of pleiotropic IVs as the primary analysis because excluding these IVs substantially reduces the number of instruments available, which in turn can compromise the precision and power of the causal estimates. Instead, we perform the MR analysis using all available IVs as our primary analysis and then use the removal of pleiotropic IVs (via MR-PRESSO and additional sensitivity tests like the MR-Egger intercept and heterogeneity tests) as a critical sensitivity analysis. This approach allows us to confirm that the observed associations remain robust even when potentially confounding variants are excluded, while acknowledging the limitations imposed by the reduced number of IVs after their removal.

In addition, Table 2 is now split into Table 2 and 3 that hopefully can convey the criteria of assessment of causal nature of the observed associations between the glycosylation traits and diseases (see Lines 1230 - 1255).

11. Line 432: *“we demonstrate, for the first time, that specific genes are associated with both N-glycosylation and liver disease, offering a starting point for genetically-guided investigation of the functional mechanisms of this phenotypic association.” I thought that this was one of the strongest statements in the manuscript and should be elaborated on and more clearly articulated throughout.*

Response: Thank you for your comment. We believe that this comment was already stated by the Reviewer in the “Remarks to the Author” paragraph and was addressed by us there. Nevertheless, here we briefly describe our response.

In the revised manuscript (see Lines 472-475), to strengthen our claims, we investigated causal relationships between all measured N-glycan levels and non-alcoholic fatty liver disease (NAFLD) —a common chronic liver disease—as well as four clinically relevant liver biomarkers: blood levels of alanine aminotransferase (ALT), alkaline phosphatase (ALP), gamma-glutamyl transferase (GGT), and aspartate aminotransferase (AST), which reflect distinct aspects of liver physiology and health.

Our MR results demonstrate that increased levels of glycans PGP25 and PGP27 (both possessing the tri-sialylated, tri-antennary A3G3S3 structure), predominantly present on alpha-1-acid glycoprotein, alpha-1-antitrypsin, and beta-2-glycoprotein 1 (primarily hepatocyte-derived proteins) [<https://doi.org/10.1016/j.eng.2023.03.013>], are significantly associated with lower AST levels. These glycoproteins play critical roles in modulating systemic inflammation, oxidative stress, and immune responses, processes directly relevant to liver injury and repair [<https://doi.org/10.4254/wjh.v13.i11.1688>]. Therefore, the association between increased A3G3S3 glycan levels and lower AST might reflect a protective effect against hepatocellular damage or stress.

We have adjusted our conclusions to reflect that while our data robustly supports a genetic association between the A3G3S3 glycan structure and AST, further studies with enhanced statistical power are necessary to clarify potential broader implications for liver health.

12. Line 451: *Can you comment on whether or how the horizontal pleiotropy test can help to distinguish between these two hypotheses?*

Response: We thank the reviewer for this comment. In our manuscript, we discuss two potential mechanisms underlying the colocalization of glyQTLs with pQTLs of N-glycoproteins. One possibility is that genetic variants may reduce overall protein levels, resulting in a corresponding decrease in the glycans associated with that protein (the first hypothesis). Alternatively, the genetic variants might directly influence N-glycoproteins' glycosylation, thereby affecting the SomaLogic assay's binding affinity and, consequently, the measured protein level (the second hypothesis). We note that these mechanisms are not mutually exclusive and may occur simultaneously. This clarification has been added to the main text (Lines 500-510).

We propose that performing bidirectional Two-Sample Mendelian randomization analyses should provide more direct evidence for distinguishing between the hypothesis that genetic variants reduce protein levels and the hypothesis that they alter glycosylation affecting assay binding. To investigate the directionality of the causal relationship between blood levels of CFH and HP and levels of their N-glycan, we conducted bidirectional Mendelian randomization analyses (Supplementary Table 16 b) coupled with MR-Egger regression to test the horizontal pleiotropy. The results support the hypothesis that changes in the blood levels of these proteins affect the levels of their N-glycans with limited evidence for horizontal pleiotropy, whereas the alternative hypothesis—where glycan levels influence protein levels—is not supported by our MR analyses.

Reviewer: 2

Remarks to the Author

1. *Sharapov et al. present the largest genome-wide association study to date of plasma protein N-glycosylation, including 117 univariate and 21 multivariate glycosylation traits and identifying 16 novel loci compared to previous studies. The authors prioritise the most likely effector genes in each loci and describe them with regard to their glycosylation patterns. They show that glyQTLs colocalize with GWAS signals for various diseases and traits, and that genetically predicted glycosylation is associated with disease outcomes in UKBB. Finally, they use bi-directional Mendelian Randomization to highlight some potentially causal associations between glycosylation traits and diseases. The work is extensive and the authors have done a good job of highlighting the novelty of their results, providing new insights into the relationships between candidate genes, N-glycosylation and disease. The methods are appropriate, descriptions are detailed and a lot of additional information is provided in supplementary data.*

Response: Thank you very much for your comments and evaluation of our submission.

Specific comments:

1. *For the gene prioritization - how was the primary gene selected if multiple genes had the same score?*

Response: Thank you for pointing out this uncertainty regarding gene prioritization. When multiple genes had the same prioritization score, we did not select the “primary” gene, but we prioritized all genes with the same score (≥ 2). However, to avoid

cluttering in tables and figures, we selected one gene (randomly) to represent the locus. In response, we have substituted the sentence (Line 712-714) “In the case of equality of the scores for two genes, we prioritized both genes” by “In the case of equality of the scores for two genes, we prioritized both genes, but we selected the random one to represent the locus in the tables/figures to avoid cluttering” in Methods section.

2. *For the colocalization with eQTLs and pQTLs, please clarify how a gene was prioritized if the glyQTL colocalized with multiple genes and proteins? For example in ST-6b, the SNP rs1260326 is annotated with eQTL NRBP1 while in ST-6f it seems to colocalize with 7 different genes. Similarly, the SNP rs8283 annotated to pQTL for C4 in ST-6b colocalizes with 6 proteins in table ST-6g. It would also be helpful to add cis/trans information to the pQTL table (ST-6g).*

Response: Thank you for your feedback. To clarify, we performed colocalization analysis with both cis and trans eQTLs/pQTLs, but in our prioritization, we considered only cis. Thus, the discrepancies between the number of prioritized genes in ST-6b and ST-6f,g are due to the omission of trans eQTLs/pQTLs from the prioritization. We have revised tables ST-6f,g to include information on whether the secondary associations are classified as cis or trans, and we have added a definition of cis in the Methods section (Line 744-746).

3. *The expression of candidate genes is shown for hepatocytes and plasma cells (Fig 2a), which is valid given established knowledge of glycosylation of plasma proteins, but did the authors perform any formal tissue specificity enrichment analysis of the GWAS results? The authors state that the results indicate strong tissue-specificity of N-glycosylation in liver and lymphoid tissue but it seems that these tissues were preselected.*

Response: Thank you for your feedback. Indeed, these tissues were preselected. Previously we have published a table summarising the average concentration and likely source tissue of major human blood plasma glycoproteins in the review (DOI: 10.1016/j.eng.2023.03.013, Table 1, Supplementary Table 1). The table was compiled using data from at least three sources - [1] average plasma concentration of the glycoprotein ($\text{mg}\cdot\text{mL}^{-1}$) and range of plasma concentration of the glycoprotein ($\text{mg}\cdot\text{mL}^{-1}$) from DOI: 10.1007/s10719-015-9626-2; [2] average concentration of the glycoprotein estimated by immunoassays ($\text{pg}\cdot\text{L}^{-1}$) and concentration of the glycoprotein estimated by MS ($\text{pg}\cdot\text{L}^{-1}$) from DOI: 10.1126/scisignal.aaz0274; [3] tissues with high expression of the gene encoding the glycoprotein as well as likely producing cells from <https://www.proteinatlas.org/humanproteome/tissue>. This data tells us that the majority of glycoproteins originate from just two tissues: lymphoid and liver. Therefore, it is logical to hypothesize that the glycosylation of the plasma proteome may be driven mainly by the specifics of the regulation of glycosylation in the lymphoid tissues (mainly plasma cells) and liver (hepatocytes).

To avoid misleading implications regarding our analysis of tissue specificity across a broad range of tissues, we renamed the paragraph 'Tissue-specific regulation of plasma protein N-glycosylation' to 'Tissue-specific regulation of plasma protein N-glycosylation in liver and lymphoid tissue' and made corresponding changes in the text.

4. *A big part of the results describes colocalization of glyQTLs with disease loci, PGS associations with diseases and MR analysis but these are not really referred to in the discussion. What could be the mechanisms at these loci? Is it likely that the disease risk is mediated through glycosylation or the other way around? The MR results indicate some bi-directional relationships that could be discussed.*

Response: Thank you for your feedback. We have revised the discussion to include the PGS and MR results (Lines 444-465):

“We found positive associations between cardiovascular disease and PGS for high-mannose glycans, previously suggested as biomarkers for cardiovascular events in patients with diabetes⁵⁷, and negative associations between metabolic disorders and PGS for galactosylated glycans. MR indicated positive causal links relationship between disorders of lipoprotein metabolism and PGS for M9 and Mtotal glycans, and between PGS for M6 glycan and asthma. Elevated high mannose glycans (e.g., M9, M6) may reflect disrupted glycan processing in the Golgi apparatus [58,59], a common feature in various pathological conditions including cancer [60]. Alternatively, their increase plasma abundance could result from elevated levels of specific carrier proteins. For instance, apolipoprotein B-100 [2], abundant in high-mannose structures (M9) [2], is closely linked to lipid disorders and cardiovascular disease [61–66], whereas elevated IgE antibodies, which prominently carry M6 glycans [2], play a role in allergic inflammation and asthma [67]”.

Additionally, we discussed the results of colocalization analysis between glyQTLs and pQTLs and their possible mechanism in the Lines (500-510).

5. *In the discussion, the authors suggest that colocalization between glyQTLs and cis-pQTLs could indicate either changes in protein levels or changes in SomaScan binding affinity due to glycosylation changes. Could the latter be even more important mechanisms when considering trans-pQTLs? Many of the colocalized loci in ST-6g are established trans pQTL hotspots observed in proteomic GWAS studies, including ABO, CHF and SERPINA1.*

Response: Thank you for your comments regarding the colocalization of glyQTLs and pQTLs. We appreciate your suggestion to consider the implications of glycosylation changes, particularly in the context of trans-pQTLs.

Firstly, as a response to another reviewer’s comment, we added summary information about the colocalization of glyQTLs with trans-pQTLs in the Results section (Lines 243-245): “Additionally, 13 glyQTLs were found to colocalize with a trans-pQTL (see Supplementary Figure 1); however, we did not include the trans-pQTL as a predictor in gene prioritization.”

Secondly, in our revised discussion (Lines 533-540), we emphasized the role of glycosylation in influencing both cis and trans effects, particularly concerning established trans-pQTL hotspots. The revised version: “However, the glyQTLs located at 15 genes, including the glycosyltransferase genes

ABO, *ST3GAL4*, *MGAT3*, and *FUT8*, are colocalized with hotspots of trans pQTLs effects [75]. This is in line with the hypothesis that technical variability in SOMALogic and Olink assays may be influenced by N-glycosylation effects on binding between proteins and probes [75]. However, our results do not conclusively prove that glycosylation alterations are the primary mechanism behind the trans-pQTL signals.”

Thirdly, to investigate the directionality of the causal relationship between blood levels of CFH and HP and levels of their N-glycan, we conducted bidirectional Mendelian randomization analyses (Supplementary Table XXX). The results support the hypothesis that changes in the blood levels of these proteins affect the levels of their N-glycans, whereas the alternative hypothesis—where glycan levels influence protein levels—is not supported by our MR analyses. We have updated the Discussion section (see Lines 460-472) with corresponding text.

6. *It would be helpful to highlight parts of the supplementary tables to link better with the findings in the main text*
- a. *The 40 novel loci could be better indicated in supplementary tables 5a-b*

Response: Thank you for your feedback. The 40 novel loci (32 identified through univariate analysis and 8 through multivariate analysis) are now clearly highlighted in bold in Supplementary Tables 3a-b for better visibility.

- b. *Lines 174-184: the text describes nine loci with secondary signals, could these be highlighted in suppl table 6a?*

Response: Thank you for your comment. The lines describing loci with secondary signals are now highlighted in bold in Supplementary Table 4 for easier identification.

- c. *Lines 225-228: it is difficult to see how the 2 colocalized proteins and 10 genes mentioned here are selected from ST 6g-f. Maybe these could be highlighted and/or the reader here also referred to ST-6b. It would also be helpful for the pQTL table if cis and trans pQTLs were indicated.*

Response: Thank you for your comment. References to ST-7a have been added to the text. Also, cis and trans indications have been added to ST-7e and ST-7f.

Minor comments:

7. *Fig1a: The scale of the Y axis is unclear. Are these $-\log_{10}(P\text{-values})$ in the outer circles? number of associations in the middle? Please clarify.*

Response: Thank you for your feedback. We have provided a more detailed description to make Figure 1a clearer and easier to understand.

8. *P.5, line 160 – Wrong table reference, should this be Supplementary Table 3b?*

Response: Thank you for pointing out this error. It has been corrected.

9. P7, line 244: HRP should be HPR

Response: Thank you for pointing out this error. It has been corrected (Lines 256).

10. Can you please specify what the columns and heatmap values in Supplementary Figure 3 reflect ? (variants and (|□| values?)

Response: Thank you for the comment. The description of the Supplementary Figure 4 was unclear and rather confusing. In the corrected version of SF4 we have clarified the description.

11. I think not all Supplementary Tables are referred to nor in the correct order

Response: Thank you very much for pointing this. We have corrected the order of the Supplementary Tables and Figures. Additionally, we combined some Supplementary Tables for convenience.

Reviewer #3 (Remarks to the Author):

Response: Thank you for participating in the revision of our manuscript.

Responses to the reviewers

Reviewer: 1

Remarks to the Author

1. All of our concerns have been addressed in these revisions.

Response: We thank the reviewer for the comments and evaluation of our submission.

Reviewer: 2

Remarks to the Author

1. Overall the authors have addressed my comments.

Response: Thank you very much for your comments and evaluation of our submission.

Additional comments:

1. However, I have a couple of additional comments regarding the newly added bi-directional MR analyses in Supplementary Tables 16a-b. First, these should be referred to in the Results and not only in the Discussion.

Response: Thank you for your suggestion. We have put our MR findings into the Results section (Lines 420-435), clearly referencing Supplementary Data 16a. Specifically, we now highlight MR associations observed for AST with five glycans—PGP27 (A3G3S3; strongest effect size, $\beta_{MR} = -0.06138$), PGP25 (also A3G3S3), PGP30 (A3F1G3S3), PGP92 (FUC-A, antennary fucosylation), and PGP113 (G4Fa/G4total). Also, we have described the MR findings in the Results section, referencing Supplementary Data 16b (Lines 228-236).

2. Second, it is difficult to link the text in Discussion to the results in ST16a without the exposure number (PGPxx) being indicated in the text or the description (A3G3S3) in the table.

Response: Thank you for your comment. We have added the description of PGPs in ST16a, ST16b.

3. The authors state that they do not find causal associations for ALP or GGT – yet such associations are shown in ST16a? Do they mean for specific exposures? This is unclear.

Response: Thank you for raising this important point. Indeed, the previous version of the manuscript lacked clarity here. Supplementary Table 16a included significant MR associations for AST, ALP, and GGT; however, only the association between A3G3S3 (PGP27 and PGP25) and AST was discussed.

In our previous submission, we (erroneously) forgot to add the results of heterogeneity tests of instrumental variables to Supplementary Table 16a. We have now added them to the revised table (Supplementary Data 16a). The heterogeneity test results show that associations observed for ALP and GGT exhibited substantial heterogeneity, indicating likely violations of MR assumptions. Therefore, we do not consider these associations as robust causal signals and have explicitly clarified this point in both the Results and Discussion sections of the revised manuscript (Lines 431-435, 470-484).

Additionally, in the previous version, we focused on discussing the association of A3G3S3 with AST, which showed the largest effect size of MR, and ignored other significant and robust associations—PGP30 (A3F1G3S3), PGP92 (FUC-A, antennary fucosylation), and PGP113 (G4Fa/G4total). In the revised version, we included these findings in the Results and Discussion sections (Lines 420-431, 470-484).

4. Please check the phrasing of results in the MR/PGS discussion – especially this sentence „MR indicated positive causal links between disorders of lipoprotein metabolism and PGS for M9 and Mtotal glycans, and between PGS for M6 glycan and asthma.“ In my opinion, it would be more appropriate to say that the MR analysis indicates a causal effect of x (M9 and Mtotal glycans) on y (lipoprotein metabolism). This should also be considered in the Results.

Response: Thank you for your edits. Following your suggestion, we have revised the sentence to: "MR analysis indicated a causal effect of M9 and Mtotal glycans on disorders of lipoprotein metabolism, and M6 glycan on asthma." (Lines 455-458). We also searched for other occurrences of this phrasing throughout the text.

5. Also, please check the spelling overall [for example “abundance”, “instance”, “links relationship”, “and. negative”].

Response: Thank you for your comment! We have checked the spelling throughout the text.

Reviewer #3 (Remarks to the Author):

Response: Thank you for participating in the revision of our manuscript.